# Organisms with alternative genetic codes resolve unassigned codons via mistranslation and ribosomal rescue

Natalie Jing Ma[1,2†], Colin F Hemez[2,3†], Karl W Barber[2,4†], Jesse Rinehart[2,4*], Farren J Isaacs[1,2*]

[1]Department of Molecular, Cellular & Developmental Biology, Yale University, New Haven, United States; [2]Systems Biology Institute, Yale University, West Haven, United States; [3]Department of Biomedical Engineering, Yale University, New Haven, United States; [4]Department of Cellular & Molecular Physiology, Yale University School of Medicine, New Haven, United States

**Abstract** Organisms possessing genetic codes with unassigned codons raise the question of how cellular machinery resolves such codons and how this could impact horizontal gene transfer. Here, we use a genomically recoded *Escherichia coli* to examine how organisms address translation at unassigned UAG codons, which obstruct propagation of UAG-containing viruses and plasmids. Using mass spectrometry, we show that recoded organisms resolve translation at unassigned UAG codons via near-cognate suppression, dramatic frameshifting from at least $-3$ to $+19$ nucleotides, and rescue by *ssrA*-encoded tmRNA, ArfA, and ArfB. We then demonstrate that deleting tmRNA restores expression of UAG-ending proteins and propagation of UAG-containing viruses and plasmids in the recoded strain, indicating that tmRNA rescue and nascent peptide degradation is the cause of impaired virus and plasmid propagation. The ubiquity of tmRNA homologs suggests that genomic recoding is a promising path for impairing horizontal gene transfer and conferring genetic isolation in diverse organisms.

DOI: https://doi.org/10.7554/eLife.34878.001

*For correspondence:
jesse.rinehart@yale.edu (JR);
farren.isaacs@yale.edu (FJI)

†These authors contributed equally to this work

Competing interests: The authors declare that no competing interests exist.

## Introduction

The standard genetic code allows faithful translation of proteins across nearly all living organisms and enables horizontally transferred genetic elements (HTGEs), such as conjugative plasmids and viruses, to exploit a host's translational machinery (*Krakauer and Jansen, 2002*). Since naturally occurring exceptions to the standard genetic code exist (*Ambrogelly et al., 2007*; *Knight et al., 2001*), researchers have hypothesized that such alternative genetic codes might arise to escape viral predation (*Shackelton and Holmes, 2008*). Recent research supports this hypothesis, with modification to codon usage or the genetic code reducing the ability of viruses and conjugative plasmids to exploit their hosts (*Coleman et al., 2008*; *Lajoie et al., 2013b*; *Ma and Isaacs, 2016*). Given the medical, technological, and evolutionary importance of HTGE-mediated horizontal gene transfer (HGT) (*Davies, 1994*; *Gogarten and Townsend, 2005*; *Moe-Behrens et al., 2013*; *Ochman et al., 2000*), understanding the molecular basis for how alternative genetic codes impede HTGEs is vital.

At the molecular level, an alternative genetic code arises from reassignment of one or more codons in the genetic code, which stems from a change in the ability of an aminoacyl-tRNA or release factor (RF) to recognize codon(s) during translation. One possible alteration of the genetic code is the loss of a codon assignment through the deletion or modification of an aminoacyl-tRNA or release factor, removing the cell's ability to decode that codon (*Figure 1A*). Such unassigned

**eLife digest** Usually, DNA passes from parent to offspring, vertically down the generations. But not always. In some cases, it can move directly from one organism to another by a process called horizontal gene transfer. In bacteria, this happens when DNA segments pass through a bacterium's cell wall, which can then be picked up by another bacterium. Because the vast majority of organisms share the same genetic code, the bacteria can read this DNA with ease, as it is in the same biological language.

Horizontal gene transfer helps bacteria adapt and evolve to their surroundings, letting them swap and share genetic information that could be useful. The process also poses a threat to human health because the DNA that bacteria share can help spread antibiotic resistance. However, some organisms use an alternative genetic code, which obstructs horizontal gene transfer. They cannot read the DNA transmitted to them, because it is in a different 'biological language'. The mechanism of how this language barrier works has been poorly understood until now.

Ma, Hemez, Barber et al. investigated this using *Escherichia coli* bacteria with an artificially alternated genetic code. In this *E. coli*, one of the three-letter DNA 'words' in the sequence is a blank – it does not exist in the bacterium's biological language. This three-letter DNA word normally corresponds to a particular protein building block. Using a technique called mass spectrometry, Ma et al. analyzed the proteins this *E. coli* forms. The results showed that it has several strategies to deal with DNA transmitted horizontally into the bacterium. One method is destroying the proteins that are half-created from the DNA, using molecules called tmRNAs. These are part of a rescue system that intervenes when protein translation stalls on the blank word. The tmRNAs help to add a tag to half-formed proteins, marking them for destruction.

This mechanism creates a 'genetic firewall' that prevents horizontal gene transfer. In organisms engineered to work from an altered genetic code, this helps to isolate them from outside interference. The findings could have applications in creating engineered bacteria that are safer for use in fields such as medicine and biofuel production.

DOI: https://doi.org/10.7554/eLife.34878.002

codons are found in alternative genetic codes in nature (*Knight et al., 2001*) and have been engineered into genomically recoded organisms (GROs) derived from *Escherichia coli* (*Isaacs et al., 2011*; *Lajoie et al., 2013b*). We recently demonstrated that a GRO with an unassigned UAG codon (i.e. lacking all instances of the UAG codon and release factor 1, RF1) impaired the propagation of HTGEs carrying UAG-ending genes, illustrating that alternative genetic codes can obstruct HGT (*Ma and Isaacs, 2016*) and establishing the GRO as an ideal model to study the molecular mechanisms that act at unassigned codons to impair HTGEs.

Encountering an unassigned codon during translation leads to ribosomal stalling, and without resolution, to cell death (*Keiler and Feaga, 2014*). However, the survival of organisms engineered to lack RF1 but retaining some UAG codons in their protein-coding sequences (*Heinemann et al., 2012*; *Mukai et al., 2010*) and the ability of GROs to resist exploitation by and continue growth in the presence of HTGEs (*Ma and Isaacs, 2016*) indicates that *E. coli* can resolve translation at unassigned UAG codons. We hypothesize that three mechanisms could resolve translation at prokaryotic ribosomes encountering these unassigned codons, each resulting in peptides with different C-terminal sequences (*Figure 1B*): (1) suppression of the codon by a near-cognate or mutated tRNA (e.g. amber suppressor) and continued translation, (2) frameshifting of bases along the mRNA transcript into a new reading frame and continued translation, or (3) stalling that elicits one of three ribosomal rescue pathways (tmRNA-SmpB, ArfA, or ArfB) in the cell (*Keiler, 2015*). The tmRNA-SmpB system acts as the primary rescue mechanism in prokaryotes, resolving ribosomal stalling that arises from the translation of mRNAs lacking a stop codon due to mRNA degradation, frameshifting, and stop codon read-through (*Keiler, 2015*). tmRNA-SmpB can also rescue ribosomes stalled on intact mRNAs for structural reasons (*Cruz-Vera et al., 2011*; *Keiler, 2015*; *Li et al., 2012*). The *ssrA*-encoded tmRNA associates with SmpB to form the tmRNA-SmpB complex, which adds a C-terminal degradation tag to peptides on stalled ribosomes (*Tu et al., 1995*). ArfA and ArfB, the secondary ribosomal rescue systems, alleviate stalling and release the stalled ribosome's nascent peptide

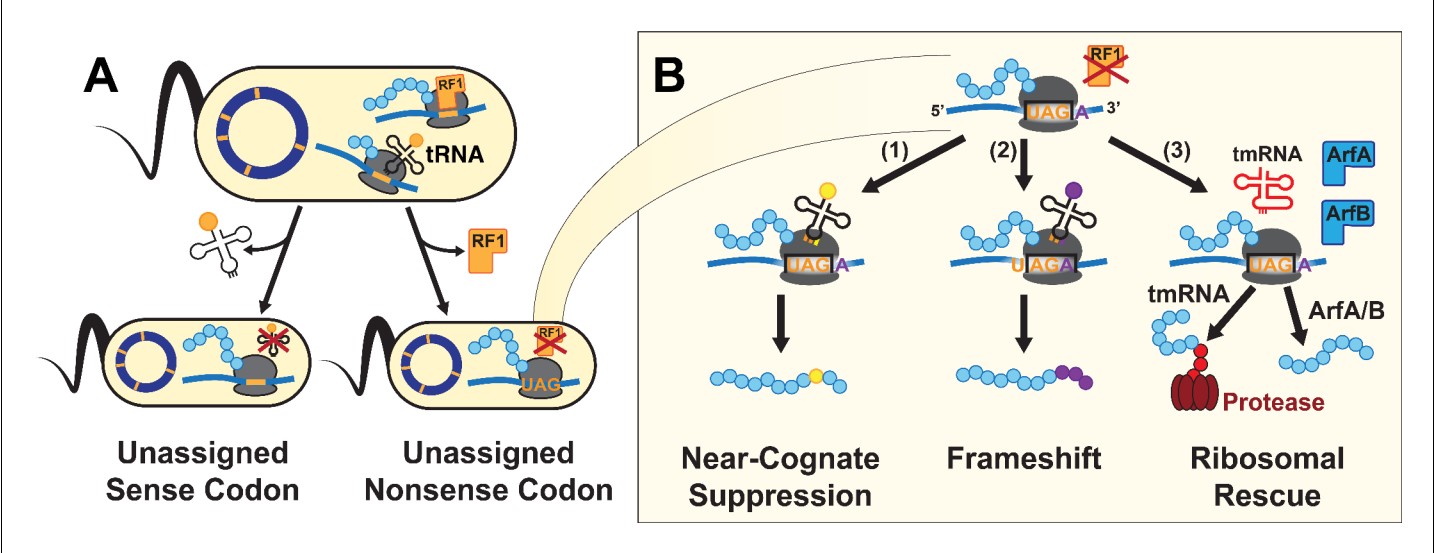

**Figure 1.** A UAG-ending transcript in the genomically recoded organism (GRO) may produce proteins with multiple differing C-termini. (**A**) Unassigned codons arise when either the cognate tRNA or release factor recognizing a codon are removed. (**B**) Since the GRO lacks Release Factor 1 (RF1), ribosomal stalling at the UAG codons results in three possible fates for the nascent protein (blue): (1) suppression of the codon by a near-cognate or suppressor tRNA (yellow) and continued translation, (2) frameshifting of bases along the mRNA transcript into a new reading frame and continued translation (purple), or (3) ribosomal rescue by the *ssrA*-encoded tmRNA, ArfA, or ArfB proteins. If ribosomal rescue occurs via tmRNA, the resulting protein is tagged with a peptide sequence (red) for degradation, while rescue via ArfA or ArfB results in release of peptide without C-terminal modification.

DOI: https://doi.org/10.7554/eLife.34878.003

without modification (*Chadani et al., 2012*; *Shimizu, 2012*). tmRNA, ArfA, and ArfB all act on non-stop ribosomal complexes, which are stalled ribosomes that have reached the 3' end of an mRNA because of stop-codon readthrough or because of the loss of a stop codon due to 3' exonuclease degradation (*Keiler, 2015*). A possible fourth outcome identified from in vitro studies is loss of translational fidelity after the ribosome encounters rare or unassigned codons (*Gingold and Pilpel, 2011*), followed by untemplated termination by release factor 2 (RF2) (*Zaher and Green, 2009*).

Studies of ribosomal stalling arising at rare codons (*Hayes et al., 2002*) or in contexts of depleted or inefficient cognate decoding elements (*George et al., 2016*; *Li et al., 2007*; *Roche and Sauer, 1999*) suggest that a number of these mechanisms could resolve translation at unassigned codons, but a lack of well-characterized model organisms with an unassigned codon has precluded direct study of this question. Here, we use the GRO as a model to demonstrate that unassigned UAG codons in mRNA transcripts (1) elicit suppression, ribosomal frameshifting, and ribosomal rescue, (2) can induce ribosomal frameshifting from at least −3 to +19 nucleotides, and (3) lead to total loss of translational fidelity. By selectively deleting ribosomal rescue pathways in the GRO, we show that the tmRNA system is primarily responsible for rescuing ribosomes stalled at unassigned codons, with deletion of the tmRNA restoring expression of UAG-ending genes and re-enabling propagation of UAG-containing plasmids and viruses in the GRO. Our work reveals mechanistic details into how cells rescue ribosomes stalled at unassigned stop codons, providing insight into how alternative genetic codes act as barriers to HTGEs and demonstrating genomic recoding as a broadly applicable strategy to obstruct HGT in engineered organisms.

## Results

### Suppression, ribosomal frameshifting, and tmRNA-mediated peptide tagging occur at unassigned codons

In prior work, we constructed an *Escherichia coli* strain in which all UAG codons were mutated to UAA, permitting the deletion of release factor 1 (RF1) and resulting in an organism that lacks a

codon assignment of UAG. This genomically recoded organism (GRO) (*Isaacs et al., 2011*; *Lajoie et al., 2013b*) exhibited resistance to multiple viruses and failure to propagate conjugative plasmids (*Lajoie et al., 2013b*; *Ma and Isaacs, 2016*) attributable to the unassigned UAG codon, but the molecular mechanisms that resolve unassigned UAG codons during translation remained unknown. In this study, we conducted two main experiments to uncover these mechanisms: (1) analysis of proteins translated from UAG-ending transcripts via mass spectrometry and western blots and (2) phenotypic assays to assess whether gene deletions of specific rescue factors restored the ability of conjugative plasmids and viruses to exploit the GRO. Since we hypothesized that the tmRNA-mediated response may resolve ribosomal stalling at the UAG codon, we also mutated the degradation tag encoded by the tmRNA from AANDENYALAA (AA-tag) to AANDENYALDD (DD-tag) for protein expression for mass spectrometry experiments. This mutation increases the half-life of protein products released by tmRNA (*Keiler et al., 1996*; *Roche and Sauer, 1999*), enabling their detection via mass spectrometry.

We assembled plasmids (pUAG-GFP and pUAA-GFP) encoding GFP genes with C-terminal 6x-His tags positioned immediately upstream of a UAG or UAA stop codon. We then expressed GFP from pUAG-GFP and pUAA-GFP in GRO cells containing the RF1-encoding *prfA* gene (GRO.DD.*prfA+*) or in GRO cells lacking *prfA* and consequently without UAG assignment (GRO.DD) (*Figure 2A*; *Table 1*; see also Key Resources Table for a list of plasmids used in this study). We then purified proteins by nickel affinity chromatography, performed trypsin digest, and used tandem mass spectrometry to collect peptide mass data as described previously (*Aerni et al., 2015*; *Amiram et al., 2015*). To distinguish between mechanisms of ribosomal rescue and mistranslation at the UAG codon, we searched mass spectrometry data with theoretical peptide libraries detailed in *Table 2* (see also *Supplementary file 3 and 4*) to identify evidence for suppression, ribosomal frameshifting, rescue via tmRNA tagging, and loss of translational fidelity.

In the GRO lacking UAG assignment, the UAG codon elicited a combination of ribosomal rescue mechanisms and mistranslation events, including tmRNA-mediated tagging, near-cognate suppression, and frameshifting. The mutated *ssrA* DD-tag appended directly to the C-terminus of GFP (LEHHHHHHAANDENYALDD) appeared in both UAG- and UAA-ending transcripts in GRO.DD and GRO.DD.*prfA+* (*Figure 2*, *Supplementary file 1* – Table S1), consistent with previous reports that overexpressed proteins are targeted for degradation by the tmRNA (*Baneyx and Mujacic, 2004*; *Li et al., 2007*; *Moore and Sauer, 2005*; *Tu et al., 1995*). Both samples also contained the unmodified C-terminus of GFP (LEHHHHHH). In GRO.DD.*prfA+*, this is likely due to translational termination via RF1, while in GRO.DD this may represent rescue of nonstop ribosomes by ArfA/ArfB, release of nascent peptides undergoing translation at the time of cell lysis, or spontaneous dissociation of the ribosome, although this last event is estimated to occur fewer than once per 100,000 codon decoding events (*Keiler and Feaga, 2014*). While these were the only C-terminal fragments detected in GRO.DD expressing UAA-GFP and in GRO.DD.*prfA+* expressing UAG-GFP, GRO.DD [pUAG-GFP] contained greater than 30 unique C-terminal sequences (*Supplementary file 2*).

The peptide fragments detected from GRO.DD [pUAG-GFP] demonstrate a combination of near-cognate suppression, ribosomal frameshifting, and tmRNA tagging (*Figure 2B*). We identified two previously known suppression events glutamine (Q) and tyrosine (Y) (*Aerni et al., 2015*; *Lajoie et al., 2013b*), and observed two new suppressors, aspartic acid (D) and valine (V). We detected ribosomal frameshifting of up to −3 (LEHHHHHHH) and +19 nucleotides (LEHHHHHHHMVR), as determined by the presence of fragments from all three reading frames appended to the C-terminal peptide of LEHHHHHH. Additionally, the LEHHHHHHHH peptide may indicate a −6 frameshift, although it is impossible to determine whether this peptide arises from a −6 frameshift or two −3 frameshifts between histidine incorporation. We also detected peptides encoded as far downstream as +82 nucleotides after the UAG codon, illustrating that the ribosome can continue translation after encountering the unassigned UAG codon provided that stalling at the UAG codon is resolved. Lastly, we identified the modified *ssrA* DD-tag at both the site of the UAG codon and downstream on multiple peptides.

Prior research in vitro revealed that a mistranslation event increases the likelihood of subsequent mistranslation events and termination by release factor 2 (RF2) (*Zaher and Green, 2009*), and we investigated whether we could detect peptides representing such mistranslation events. Given the difficulty of distinguishing such peptides from suppression or frameshifting with one or two amino

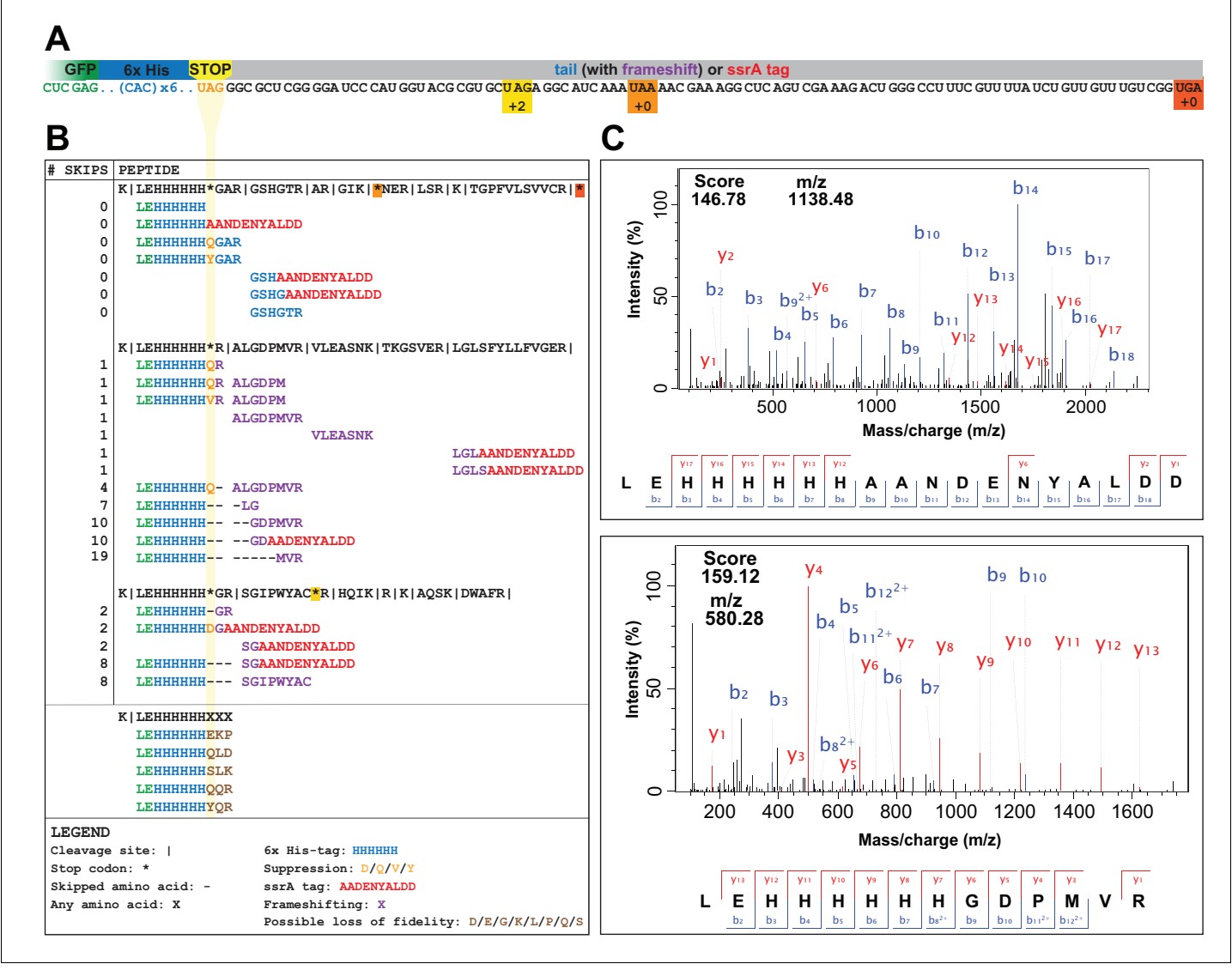

**Figure 2.** UAG codons in the genomically recoded organism elicit suppression, frameshifting, and tagging for degradation by the tmRNA. (**A**) Schematic of the GFP construct with a C-terminal 6x-His tag and a UAG stop codon, showing 102 nucleotides downstream of the UAG codon and the positions of other stop codons in the downstream tail. (**B**) Peptides identified from the C-terminus of a UAG-ending GFP construct expressed in the GRO (using libraries detailed in *Supplementary file 3 and 4*). Purified GFP protein was digested with trypsin, processed via MS/MS, and the resulting data were computationally searched using libraries encoding all possible suppressors and all possible subsequent reading frames. Peptides are mapped to the C-terminus of the original GFP construct and grouped by reading frame, with the number of bases skipped listed in the left column. Green text represents GFP, blue text represents the C-terminal 6xHis tag and unframeshifted readthrough, orange text represents the position of a UAG stop codon, purple text represents frameshifted readthrough, and red text represents the tmRNA tag. Black dashes represent ribosomal frameshifts (*Figure 2—source datas 1* and *2*). (**C**) MS-MS spectra for two peptides: the C-terminus of GFP with the appended degradation tag (LEHHHHHHAANDENYALDD) and the C-terminus of GFP demonstrating a + 10 base skip in translation (LEHHHHHHGDPMVR). The other spectra validated from UAG-GFP expressing GRO.AA are shown in *Supplementary file 2*.

DOI: https://doi.org/10.7554/eLife.34878.004

The following source data is available for figure 2:

**Source data 1.** Raw data and analysis of peptides detected in mass spectrometry datasets using a library generated to search for frameshifting, near-cognate suppression, and ribosomal rescue events (*Supplementary file 3*).
DOI: https://doi.org/10.7554/eLife.34878.005

**Source data 2.** Raw data and analysis of peptides detected in mass spectrometry datasets using a library generated to search for loss of translational fidelity (*Supplementary file 4*).
DOI: https://doi.org/10.7554/eLife.34878.006

**Table 1.** Strains used in this study.

| Strain Abbreviation* | Ancestor (source)† | Genotype | # UAG Codons‡ | RF1 Status§ | Ribosomal rescue gene deletion | ssrA tag Status# | Investigated in |
|---|---|---|---|---|---|---|---|
| GRO.DD.prfA + | GRO.AA (this study) | ΔmutS:zeo.Δ(ybhB-bioAB):[λcI857.Δ(cro-ea59):tetR-bla] | 0 | +RF1 | n/a | DD | GFP expression for mass spectrometry (*Figure 2*) |
| GRO.DD | GRO.AA (this study) | ΔmutS:zeo.Δ(ybhB-bioAB):[λcI857.Δ(cro-ea59):tetR-bla], ΔprfA, ΔtolC | 0 | ΔRF1 | n/a | DD | GFP expression for mass spectrometry (*Figure 2*) |
| ECNR2.AA | E. coli MG1655 (*Wang et al., 2009*) | MG1655 ΔmutS:zeo.Δ(ybhB-bioAB):[λcI857.Δ(cro-ea59):tetR-bla] | 321 | +RF1 | n/a | AA | Fitness, conjugation, and viral infection (*Figures 3* and *4*) |
| GRO.AA | ECNR2.AA (*Lajoie et al., 2013b*) | ΔmutS:zeo.Δ(ybhB-bioAB):[λcI857.Δ(cro-ea59):tetR-bla], ΔprfA, ΔtolC | 0 | ΔRF1 | n/a | AA | Fitness, conjugation, and viral infection (*Figures 3* and *4*) |
| GRO.AA.ΔssrA | GRO.AA (this study) | ΔmutS:zeo.Δ(ybhB-bioAB):[λcI857.Δ(cro-ea59):tetR-bla], ΔprfA, ΔtolC | 0 | ΔRF1 | ΔssrA | AA | Fitness, conjugation, and viral infection (*Figures 3* and *4*) |
| GRO.AA.ΔarfA | GRO.AA (this study) | ΔmutS:zeo.Δ(ybhB-bioAB):[λcI857.Δ(cro-ea59):tetR-bla], ΔprfA, ΔtolC | 0 | ΔRF1 | ΔarfA | AA | Fitness, conjugation, and viral infection (*Figures 3* and *4*) |
| GRO.AA.ΔarfB | GRO.AA (this study) | ΔmutS:zeo.Δ(ybhB-bioAB):[λcI857.Δ(cro-ea59):tetR-bla], ΔprfA, ΔtolC | 0 | ΔRF1 | ΔarfB | AA | Fitness, conjugation, and viral infection (*Figures 3* and *4*) |
| GRO.AA.ΔssrA.ΔarfB | GRO.AA (this study) | ΔmutS:zeo.Δ(ybhB-bioAB):[λcI857.Δ(cro-ea59):tetR-bla], ΔprfA, ΔtolC | 0 | ΔRF1 | ΔssrA, ΔarfB | AA | Fitness, conjugation, and viral infection (*Figures 3* and *4*) |
| GRO.AA.ΔarfA. ΔarfB | GRO.AA (this study) | ΔmutS:zeo.Δ(ybhB-bioAB):[λcI857.Δ(cro-ea59):tetR-bla], ΔprfA, ΔtolC | 0 | ΔRF1 | ΔarfA, ΔarfB | AA | Fitness, conjugation, and viral infection (*Figures 3* and *4*) |

*All strains derived from ECNR2, as described in **Wang et al. (2009)**.

†See **Key Resources Table** for additional information on strains and sources. The GenBank accession number for *E. coli* MG1655 is U00096, and the GenBank accession number for GRO.AA is CP006698.

‡ Out of a total of 321 in the original ECNR2 strain.

§RF1 terminates translation at UAG and UAA. Deletion of RF1 eliminates recognition of UAG during translation; translational termination continues through RF2, which recognizes UAA and UGA.

#The *ssrA* gene encodes the tmRNA, which appends the *ssrA* degradation tag to stalled ribosomes. The wild-type sequence is AANDENYAL<u>AA</u>; mutation of the C-terminus to AANDENYAL<u>DD</u> slows degradation of peptides to enable detection by mass spectrometry.

DOI: https://doi.org/10.7554/eLife.34878.007

**Table 2.** Components of peptide library constructed to search and analyze tandem mass spectrometry data.
The LEHHHHHHXXX library was separate from the library that contained the entries of the first three rows of the table (see **Supplementary file 3 and 4**).

| Library component | Example peptides (from *Figure 2A*) | Enables detection of… | Complete peptide list |
|---|---|---|---|
| Any one of 20 canonical amino acids inserted at the UAG codon | LEHHHHHH**Q**GAR | Near-cognate suppression | *Supplementary file 3* |
| Any length of C-tail following UAG codon to the next non-UAG stop codon or to 38 amino acids downstream of the UAG codon, whichever came first | ALGDPMVR | Readthrough, frameshifting, and rescue by ArfA or ArfB | *Supplementary file 3* |
| AANDENYALDD degradation tag | LEHHHHHHGD**AANDENYALDD** | Rescue by tmRNA-SmpB | *Supplementary file 3* |
| All peptides of form LEHHHHHHXXX, where X is any amino acid | LEHHHHHH**QLD** | Loss of translational fidelity | *Supplementary file 4* |

DOI: https://doi.org/10.7554/eLife.34878.008

acids, we created a hypothetical peptide library (*Supplementary file 1* – Table S2) containing all combinations of LEHHHHHHXXX, wherein X is any amino acid incorporated at the three residue positions directly downstream of the UAG codon (*Supplementary file 4*). The search with this library returned 23 unique peptides, 14 of which met our scoring threshold of 15 (*Aerni et al., 2015*). Five of these peptides (LEHHHHHHEKP, LEHHHHHHQLD, LEHHHHHHQQR, LEHHHHHHSLK, and LEHHHHHHYQR) could only arise from the mRNA transcript through two or more frameshift events after stalling at the UAG codon had already resolved (*Supplementary file 1* – Table S2), suggesting they instead arise from loss of translational fidelity and spontaneous termination of translation following mistranslation at the UAG codon. We also had enough resolution in the data to manually verify the amino acid sequences of LEHHHHHHQQR and LEHHHHHHYQR, noting a 35 Da shift in mass between the Q and Y in the third position from the C-terminus.

Although several alternative hypotheses may explain these random tripeptides, these explanations are either incomplete or unlikely given our current understanding of prokaryotic translation. First, it is improbable that these fragments arose from routine errors in mRNA transcription because this would require at least two transcriptional errors in a nine-nucleotide span. The transcription error rate in *E. coli* is estimated to be ~1 in 10,000 bases (*Blank et al., 1986*; *Rosenberger and Hilton, 1983*) and our strains have no known mutations that would lead to greater error rates in transcription. Second, it is possible that ArfA or ArfB may have terminated translation in these peptides due to 3' exonuclease shortening of the mRNA transcript as the ribosome is stalled at the UAG codon (*Keiler and Feaga, 2014*; *Yamamoto et al., 2003*). However, this does not explain the non-encoded tripeptides appended to the LEHHHHHH peptide. Lastly, the peptides LEHHHHHHQQR, LEHHHHHHSLK, and LEHHHHHHYQR may have been part of longer peptides that were cleaved off during trypsin digest. In this case, translation may have continued past the C-terminal R or K observed in these peptides, but this consideration would not apply to LEHHHHHHEKP and LEHHHHHHQLD and again does not explain the non-encoded tripeptide sequence observed appended to LEHHHHHH. Given this, we hypothesize that these five peptides result from loss of translational fidelity after stalling at the UAG codon that may lead to (1) spontaneous termination of translation due to the untemplated action of RF2 following mistranslation or (2) ArfA- or ArfB-mediated release predicated on 3' exonuclease degradation of the mRNA. The rare event of spontaneous hydrolysis of the peptide from the ribosome is also possible.

## *ssrA* and *arfB* mediate degradation of proteins containing unassigned UAG codons

Since mass spectrometry data indicated that a combination of mechanisms could resolve stalled translation at the unassigned UAG codon, we generated targeted deletions of the ribosomal rescue systems (*ssrA*, *arfA*, and *arfB*) in strains with wild-type *ssrA* sequence (GRO.AA) to determine whether protein production from UAG-ending transcripts in ΔRF1 cells could be restored to levels seen in +RF1 cells. Using recombineering (*Sharan et al., 2009*), we produced single and double deletions of the *ssrA*, *arfA*, and *arfB* genes that encode the ribosomal rescue systems. Efforts to generate a double deletion of *ssrA* and *arfA* failed (data not shown) because the resulting phenotype is synthetic lethal (*Chadani et al., 2010*). We transformed each deletion strain with the UAG-GFP construct under a highly expressing, inducible pLtetO promoter (*Lutz and Bujard, 1997*) and induced GFP expression for 20 hr, measuring the effect of protein expression on cellular growth through doubling time and maximum optical density at 600 nm ($OD_{600}$) (*Figure 3A and B*, *Supplementary file 1* – Table S3). To quantify protein expression, we then assayed whole-cell lysate from equal cell numbers, as determined by $OD_{600}$, for abundance of protein via anti-GFP western blot alongside GFP standards of known concentration as described previously (*Figure 3C*, *Figure 3— source data 6*) (*Pirman et al., 2015*). We also included as positive controls (1) a wild-type strain (ECNR2) expressing the UAG-GFP construct and (2) GRO.AA expressing UAA-GFP.

Expression of UAG-GFP impaired GRO growth rate and cell density, generating a 54% increase in doubling time and 8% reduction in maximum $OD_{600}$ compared to cells not expressing UAG-GFP, and a 25% greater doubling time and 14% lower maximum $OD_{600}$ compared to cells expressing UAA-GFP. In contrast, ECNR2 exhibited only a 7% increase in doubling time and a 5% reduction in maximum $OD_{600}$ when expressing UAG-GFP. Although deletion strains experienced reduced growth rate as measured by doubling time compared to the GRO.AA, they exhibited a less pronounced increase in doubling time when expressing UAG-GFP (increases in doubling time between 12% and

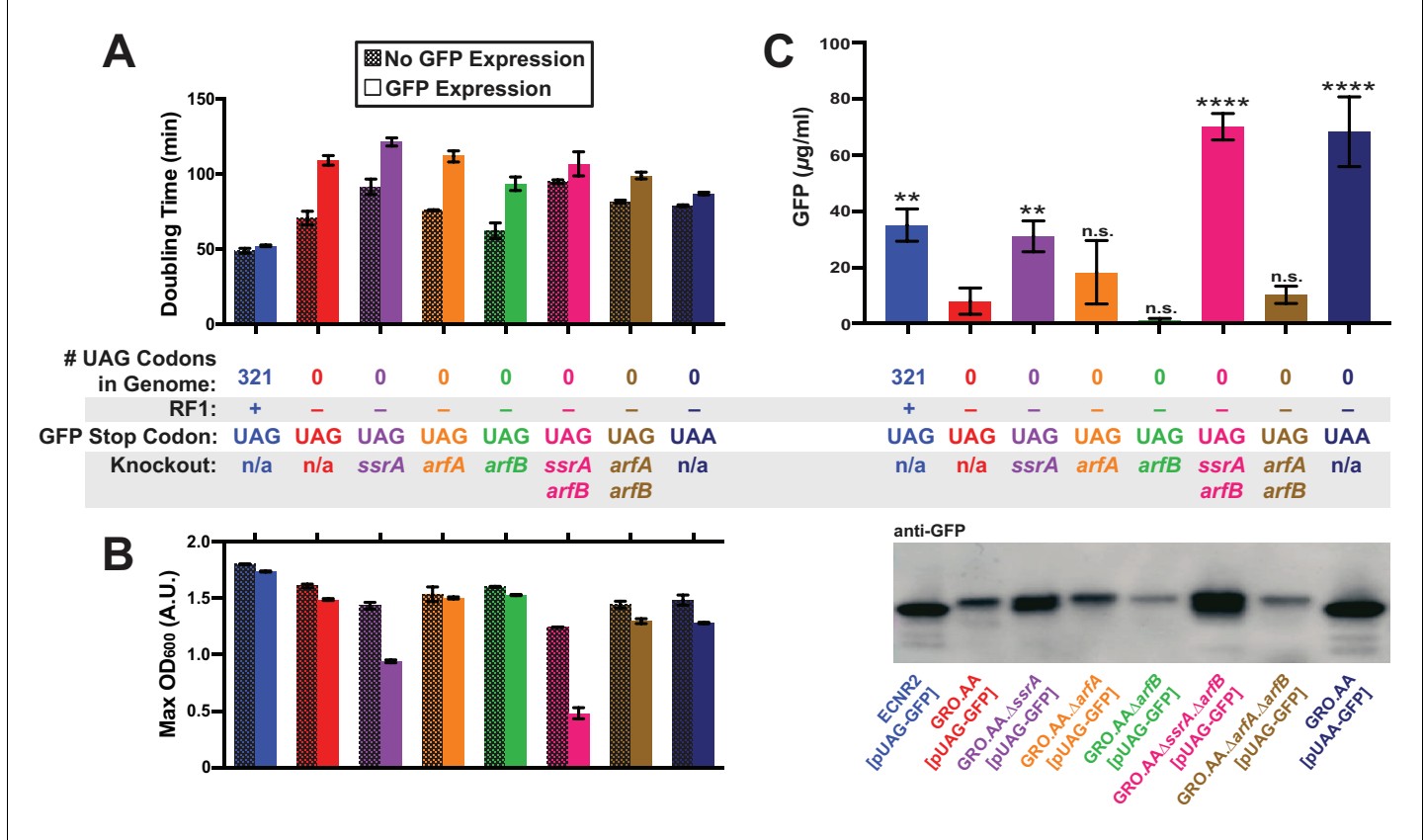

**Figure 3.** Deletion of both *ssrA* and *arfB* restores protein production in the genomically recoded organism. (**A**) Comparison of doubling times for WT and GRO strains carrying listed deletions with and without GFP induction. Error bars show standard deviation centered at mean, n = 3; data were analyzed using *Source code 1* (*Figure 3—source datas 1* and *2*). (**B**) Change in maximum optical density at 600 nm ($OD_{600}$) due to expression of UAG-GFP or UAA-GFP in wild-type (WT) and GRO strains carrying listed deletions. Error bars show standard deviation centered at mean, n = 3 (*Figure 3—source datas 1* and *2*). (**C**) Quantification of GFP abundance per 1 mL of cells at $OD_{600}$ of 2.5 via western blot from biological replicates of indicated strains (*Figure 3—source datas 3–6*). Error bars show standard deviation centered at mean, n = 3 (*Figure 3—source datas 3–5*). See *Figure 3—figure supplement 1* for linear calibration curves used to quantify GFP abundance for each replicate experiment. Image of representative western blot is below the graph. p-values are calculated in relation to the GRO containing the UAG-ending GFP (GRO – UAG) and are as follows: * is p≤0.05, ** is p≤0.01, *** is p≤0.001, and **** is p≤0.0001.
DOI: https://doi.org/10.7554/eLife.34878.009

The following source data and figure supplement are available for figure 3:

**Source data 1.** Growth curve data from 96-well plate assay analyzed using *Source code 1* (one of three plate replicates), used for data represented in *Figure 3A and B*.
DOI: https://doi.org/10.7554/eLife.34878.011

**Source data 2.** Analysis of doubling times and maximum $OD_{600}$'s of indicated strains.
DOI: https://doi.org/10.7554/eLife.34878.012

**Source data 3.** Anti-GFP western blot image used for quantification of GFP yields; replicate 1.
DOI: https://doi.org/10.7554/eLife.34878.013

**Source data 4.** Anti-GFP western blot image used for quantification of GFP yields; replicate 2.
DOI: https://doi.org/10.7554/eLife.34878.014

**Source data 5.** Anti-GFP western blot image used for quantification of GFP yields; replicate 3.
DOI: https://doi.org/10.7554/eLife.34878.015

**Source data 6.** Analysis of western blot data represented in *Figure 3C*.
DOI: https://doi.org/10.7554/eLife.34878.016

**Figure supplement 1.** Calibration curves used for quantification of GFP yields, as represented in *Figure 3C*, using GFP samples of known concentration.
DOI: https://doi.org/10.7554/eLife.34878.010

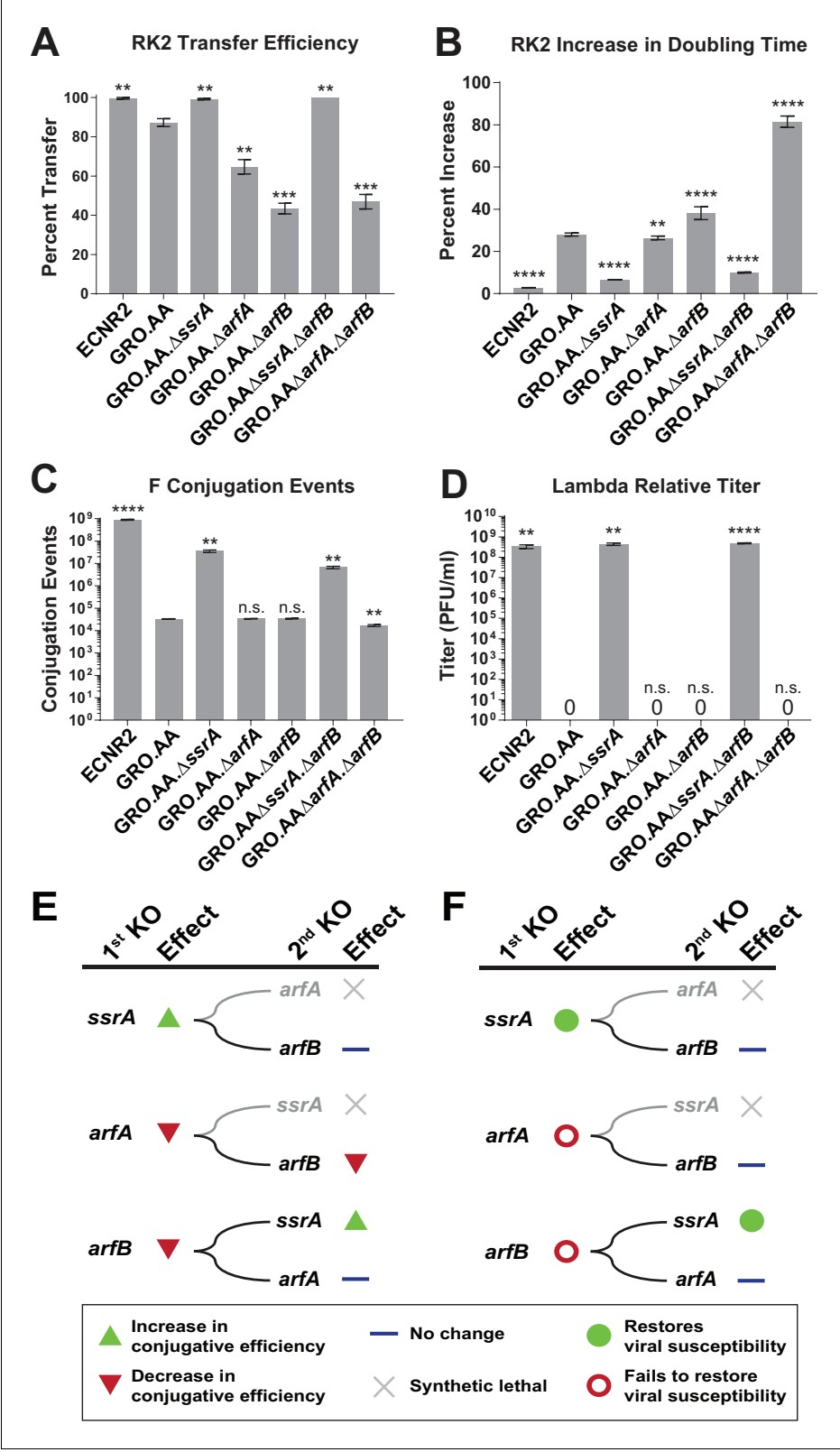

**Figure 4.** Deleting *ssrA* restores propagation of both viruses and conjugative plasmids in the genomically recoded organism. (A) Percent transfer of conjugative plasmid RK2 from a wild-type donor into wild-type (WT), GRO, or GRO with designated deletions (KO) as recipients (*Figure 4—source data 1*). Data are obtained from technical triplicates generated from a single biological sample. (B) Percent increase in doubling time for strains carrying plasmid RK2 compared to strains lacking RK2 (*Figure 4—source datas 2* and *3*). (C) Number of conjugation events for conjugative plasmid F

*Figure 4 continued on next page*

Figure 4 continued

from wild-type, GRO, or GRO with designated gene deletions as donors to a wild-type recipient (*Figure 4—source data 4*). Data are obtained from technical triplicates generated from a single biological sample. (D) Relative titer on wild-type, GRO, and GRO with designated deletions of phage λ (*Figure 4—source data 5*). Error bars show standard deviation centered at mean, n = 3. p-values are calculated in relation to the GRO condition and are as follows: * is $p \leq 0.05$, ** is $p \leq 0.01$, *** is $p \leq 0.001$, and **** is $p \leq 0.0001$. (E) Effects of sequential deletions of ribosomal rescue mechanisms on conjugative plasmid transfer efficiency. (F) Effects of sequential deletions of ribosomal rescue mechanisms on viral susceptibility.
DOI: https://doi.org/10.7554/eLife.34878.017

The following source data is available for figure 4:

**Source data 1.** Analysis of RK2 plasmid conjugation data represented in *Figure 4A*.
DOI: https://doi.org/10.7554/eLife.34878.018
**Source data 2.** Growth curve data from 96-well plate assay analyzed using *Source code 1*, used for data represented in *Figure 4B*.
DOI: https://doi.org/10.7554/eLife.34878.019
**Source data 3.** Analysis of doubling times represented in *Figure 4B*.
DOI: https://doi.org/10.7554/eLife.34878.020
**Source data 4.** Analysis of F plasmid conjugation data represented in *Figure 4C*.
DOI: https://doi.org/10.7554/eLife.34878.021
**Source data 5.** Analysis of lambda phage infection data represented in *Figure 4D*.
DOI: https://doi.org/10.7554/eLife.34878.022

50%) as compared to the GRO.AA (54% increase in doubling time) (*Figure 3A*). However, deletion of *ssrA* reduced fitness during protein expression as measured by maximum $OD_{600}$, with GRO.AA.Δ*ssrA* demonstrating a 34% reduction in max $OD_{600}$ and GRO.AA.Δ*ssrA*.Δ*arfB* demonstrating a 61% decrease in max $OD_{600}$. This is potentially due to increased presence of misfolded or prematurely truncated peptides that are ordinarily tagged and degraded by the tmRNA. Interestingly, deletion of *arfB* produces a 50% increase in doubling time during protein expression, suggesting ArfB may play a role in ribosomal rescue during high levels of ribosomal stalling.

We then investigated the impact of unassigned codons on protein production using western blot densitometry, and found that the GRO expressing UAG-GFP produced less than one-fourth of the protein amount than does ECNR2 expressing UAG-GFP (*Figure 3C*, 8.0 µg/ml for the GRO versus 35 µg/ml for ECNR2, p=0.0014). GRO.AA expressing UAA-GFP produced nearly nine times more protein than did GRO.AA expressing UAG-GFP (68 µg/ml for GRO.AA [pUAA-GFP] versus 8.0 µg/ml for GRO.AA [pUAG-GFP], p<0.0001), indicating that the UAG codon in pUAG-GFP is the cause of reduced protein expression in the GRO. Deletion of *ssrA* in the UAG-GFP-expressing GRO partially restored protein production to levels seen in its UAA-GFP-expressing counterpart with no knockouts (31 µg/ml for GRO.AA.Δ*ssrA* [pUAG-GFP] versus 68 µg/ml for GRO.AA [pUAA-GFP]) and deletion of both *ssrA* and *arfB* fully restored protein production (70. µg/ml). These *ssrA* deletion strains likely demonstrate increased GFP expression and reduced growth rate (*Figure 3A*) and cell density (*Figure 3B*) because translation of GFP transcripts sequesters cellular resources at the expense of cellular replication, producing GFP peptides that are freed from nonstop ribosomes via ArfA or ArfB without addition of a degradation tag.

A deletion of *arfB* leads to strikingly low- protein abundances from UAG-GFP transcripts that approach the lower limit of detection of our assay, although this apparent reduction in protein production was not statistically significant in comparison to protein production by GRO.AA [pUAG-GFP]. These ArfB deletion data, together with the fitness reduction observed in the GRO, suggest that ArfB is constitutively expressed and relieving low levels of ribosomal stalling in *E. coli*. These data also suggest that while deletion of *ssrA* partially recovers protein production from UAG-ending transcripts in the GRO, deletion of both *ssrA* and *arfB* is necessary to fully recover protein expression from UAG-ending transcripts to levels seen from the translation of UAA-ending transcripts in the GRO.

## Deletion of *ssrA* restores conjugative plasmid propagation and viral infection in the GRO

To determine whether deletions of of *ssrA* or *arfB* could restore propagation of horizontally-transferred genetic elements in the GRO, we assessed conjugation efficiency and growth rate from plasmids RK2 and F on GRO strains with single and double deletions of *ssrA*, *arfA*, and *arfB*. Previous

research indicates that the UAG stop codon in the *trfA* gene on RK2 leads to impaired conjugation efficiency and replication in the GRO (*Ma and Isaacs, 2016*), likely because the TrfA protein is required to initiate plasmid replication (*Pansegrau et al., 1994*). Phenotypically, this manifests as reduced efficiency of plasmid transfer in conjugation experiments and increased doubling times for RK2$^+$ strains in media selecting for plasmid maintenance due to loss of plasmid and concomitant antibiotic resistance genes. We found that deletion of *ssrA* increased the ability of the GRO to both receive (*Figure 4A*, *Supplementary file 1* – Table S4) and replicate RK2 (*Figure 4B*, *Supplementary file 1* – Table S5). RK2 conjugation efficiency in GRO.AA.Δ*ssrA* improved to 99% (compared to 87% in GRO.AA), and the strain showed an increase in doubling time of only 6% compared to a 28% increase for GRO.AA (p<0.0001). We observed similar results for GRO.AA.Δ*ssrA*.Δ*arfB*. However, single deletion of *arfB* halved RK2 conjugative efficiency (*Figure 4A*, p=0.0002). This strain also exhibited a 38% increase in doubling time when bearing RK2, compared to the 28% increase in doubling time seen in the GRO with no ribosomal rescue gene deletions (*Figure 4B*, p<0.0001).

For plasmid F (*Figure 4C*, *Supplementary file 1* – Table S6), which contains UAG-ending genes *traY* and *traL* that are essential for conjugation between cells (*Ma and Isaacs, 2016*), we found that deletion of *ssrA* increased conjugation events from the GRO donor 1,000-fold to $3.56 \times 10^7$ (p=0.0015) compared to GRO.AA ($3.30 \times 10^4$ events), *arfA* deletion ($3.41 \times 10^4$ events), and *arfB* deletion ($3.47 \times 10^4$ events). GRO.AA.Δ*ssrA*.Δ*arfB* and GRO.AA.Δ*arfA*.Δ*arfB* exhibited 5.2- and 2.3-fold decrease in conjugative efficiency when compared to GRO.AA.Δ*ssrA* and GRO.AA.Δ*arfA* single deletion strains, respectively (p<0.01 for each, *Figure 4C*). These reductions in RK2 and F conjugative efficiency attributable to *arfB* deletion indicate that ArfB likely contributes to relief of nonstop ribosomes when encoded in its native ribosomal context, supporting evidence of ArfB's ribosomal rescue activity previously validated in vitro (*Handa et al., 2011*) and when over-expressed in the absence of *ssrA* and *arfA* in vivo (*Chadani et al., 2010*). However, deletion of *ssrA* is sufficient to restore both conjugation and propagation of RK2 and F in the GRO. We next attempted infection with phage λ on our suite of deletion strains (*Figure 4D*, *Supplementary file 1* – Table S7). Although deletion of *arfA* or *arfB* does not recover viral infection, deletion of the *ssrA* gene—either alone (p=0.0016) or alongside deletion of *arfB* (p<0.0001)—recovers λ infection of the GRO to levels similar to wild-type, with about $10^8$ plaque forming units per mL (PFU/mL) (*Figure 4D*). These results demonstrate that removal of *ssrA* has the greatest influence in restoring conjugative plasmid transfer efficiency and viral susceptibility in the GRO (*Figure 4E and F*).

## Discussion

In this study, we use a genomically recoded organism (GRO) containing an unassigned UAG codon as a model to investigate the molecular mechanisms that obstruct the propagation of HTGEs in organisms with alternative genetic codes. We demonstrate that unassigned stop codons elicit near-cognate suppression, frameshifting, and the action of ribosomal rescue mechanisms (*Figure 2*). tmRNA-mediated ribosomal rescue prompted by the unassigned codon results in the degradation of nascent peptides translated from UAG-ending transcripts and obstructs the propagation of HTGEs (*Figure 3*, *Figure 4*). Additionally, *ssrA* deletion strains exhibit both significantly increased UAG-GFP yields (*Figure 3C*) and recovered propagation of HTGEs (*Figure 4*), consistent with evidence that deletion of *ssrA* removes inhibition of ArfA production and releases nascent peptides from stalled ribosomes without degradation (*Chadani et al., 2011*; *Garza-Sánchez et al., 2011*; *Schaub et al., 2012*). Our GRO model thus sheds light on the functional significance of previously described regulatory relationships while elucidating the unique mechanistic contributions of different ribosomal rescue systems in resolving translation at unassigned stop codons. These mechanistic outcomes that occur as a consequence of ribosomal stalling could be further investigated via ribosomal profiling in future work.

The mass spectrometry data collected from our GRO model demonstrate the striking proclivity for the ribosome to undergo un-programmed frameshifting at unassigned stop codons and represents, to our knowledge, the first in vivo study to examine such frameshifting. Prior studies have revealed programmed ribosomal frameshifting from −4 to +50 nucleotides (*Atkins et al., 2016*; *Baranov et al., 2015*; *Huang et al., 1988*; *Yan et al., 2015*), but these studies focused on frameshifts programmed into mRNA transcripts through combinations of four mechanisms: (1) use of

rare codons to slow translation speed at the skip site, (2) weak base pairing of the P-site tRNA anticodon and mRNA codon, (3) strong base pairing of the P-site tRNA anticodon to the location where the ribosome will re-bind the mRNA, and (4) a region six bases upstream of the re-binding site that mimics a Shine-Dalgarno sequence and offsets the energetic cost of frameshifting (*Pech et al., 2010*). Although the UAG codon in our GFP transcript slows translation, the P-site codon-anticodon pair for the codon immediately upstream of UAG is exact (CAC codon and $^{GUG}$His-tRNA anticodon) (*Hsu et al., 1984*) and any frameshift except backward would incur greater mispairing between the P site codon and anticodon. Additionally, no Shine Dalgarno-like sequence (AGGAGG) (*Shine and Dalgarno, 1974*; *Vimberg et al., 2007*) exists upstream, suggesting that the GFP construct we use contains only one of the four elements required for programmed ribosomal frameshifting (*Supplementary file 1*). From our construct, we observed frameshifts of potentially up to −6 and +19 nucleotides in response to the unassigned UAG codon (*Figure 2*, *Supplementary file 1* – Tables S1 and S2). Collectively, our work uncovers a wide variety of frameshifting events that can occur in response to ribosomal stalling in vivo, highlighting the capacity of the ribosome to continue translation despite missing an essential translational component.

Mass spectrometry analysis also revealed truncated mistranslation products that possibly represent loss of translational fidelity and termination by RF2 downstream of an initial mistranslation event at the UAG codon, known as post-peptidyl transfer quality control (*Petropoulos et al., 2014*; *Zaher and Green, 2009*), a result previously only observed in vitro. Although prior studies decades ago revealed premature truncation products in vivo (*Manley, 1978*), they lacked the technical capability to determine whether these peptides arose from a single mistranslation event or demonstrated loss of translational fidelity after the ribosome encounters a rare or unassigned codon. The mistranslation products we detect show repeated mistranslation events that could not have been produced by suppression, ribosomal rescue, or frameshifting, unless the ribosome frameshifted multiple times after resolving stalling at the UAG codon (*Figure 2B*, *Supplementary file 1*). These events may be followed by ribosomal rescue via ArfA or ArfB, spontaneous ribosomal dissociation, or termination via release factor 2, though our technique was not capable of distinguishing between these fates. Previous in vitro studies using purified ribosome complexes determined that a mistranslation event destabilized the P-site helix, reducing the ability of the A-site to discriminate between anticodons and resulting in further mistranslation events and rapid termination by RF2 with the assistance of release factor 3 (*Zaher and Green, 2009*; *Zaher and Green, 2010*). The researchers predicted that a single mistranslation event would also lead to prematurely truncated peptides with two or three miscoded C-terminal amino acids appended in vivo (*Zaher and Green, 2009*). These findings, together with our results, motivate future work to investigate the possibility of loss of translational fidelity after an initial translation error and highlight the GRO as a model for elucidating translational fidelity in vivo.

The GRO demonstrates that general ribosomal rescue mechanisms resolve ribosomal stalling at unassigned stop codons. As most sequenced bacterial species contain a homolog of the tmRNA, ArfA, or ArfB ribosomal rescue systems (*Hudson et al., 2014*; *Keiler, 2015*) and eukaryotic cells contain analogous pathways that rescue stalled ribosomes (*Graille and Séraphin, 2012*), we anticipate that translational stalling at unassigned codons can be resolved similarly in these organisms. Accordingly, we hypothesize that organisms beyond *E. coli* should tolerate unassigned codons as intermediates toward codon reassignments in genomic recoding, efforts for which are underway in numerous prokaryotic and eukaryotic species (*Lau et al., 2017*; *Napolitano et al., 2016*; *Ostrov et al., 2016*; *Richardson et al., 2017*). Additional barriers to codon reassignment exist, such as regulatory roles of codons in gene expression (*Lajoie et al., 2013a*), but our findings indicate that unassigned codons are tolerable in the absence of specialized translational machinery to address them, both as intermediate steps towards codon reassignment and as permanent parts of the genetic code.

Our findings suggest that we can use unassigned codons to engineer organisms with broad resistance to HTGEs and impart genetic isolation, increasing engineered organisms' stability in biotechnology applications. Since tmRNA homologs are found in >99% of all sequenced bacterial genomes (*Hudson et al., 2014*; *Keiler, 2015*), we would expect other organisms engineered to contain unassigned codons to exhibit immunity to horizontally transferred genetic elements. As researchers pursue further efforts in whole genome recoding (*Boeke et al., 2016*; *Lau et al.,*

*2017*; *Napolitano et al., 2016*; *Ostrov et al., 2016*; *Richardson et al., 2017*) and engineer organisms for use in open environments, we require strategies to genetically isolate such organisms from their surrounding environment to ensure robust function, both individually (*Moe-Behrens et al., 2013*) and as members of microbial communities (*Grosskopf and Soyer, 2014*; *Hillesland and Stahl, 2010*). Genomically recoded organisms with unassigned codons would possess reduced susceptibility to exploitation by HTGEs, increasing their stability in open environments. Although this work demonstrates that an unassigned stop codon acts as a barrier to HGT, this current barrier can be breached by mutation or deletion of the tmRNA to produce a functional protein. In contrast, we expect that an organism with an unassigned sense codon would have even greater barriers to HGT, as premature termination at an unassigned sense codon would likely produce a nonfunctional, truncated peptide. We thus anticipate that further genomic recoding to engineer additional unassigned sense and nonsense codons may be a broadly applicable strategy to confer genetic isolation in living systems, facilitating the safe use of engineered organisms in complex open environments.

# Materials and methods

**Key resources table** Genetic reagents, bacterial strains, antibodies, and software used in this study.

| Reagent type (species) or resource | Designation | Source or reference | Identifiers | Additional information | Isaacs Lab Reference # | Full genotype of strains | # UAG Codons | RF1 status | Ribosomal rescue gene knockout | ssrA tag status |
|---|---|---|---|---|---|---|---|---|---|---|
| Gene (*Escherichia coli*) | pUAG-GFP | this paper | eGFP-6xHis-UAG; Plasmid NJM88; Strain NJM1242 | eGFP protein with a C-terminal 6-His tag for protein purification, terminating translation in a UAG codon. | Plasmid NJM88; Strain NJM1242 | N/A | N/A | N/A | N/A | N/A |
| Gene (*E. coli*) | pUAA-GFP | this paper | eGFP-6xHis-UAA; Plasmid NJM89; Strain NJM1249 | eGFP protein with a C-terminal 6-His tag for protein purification, terminating translation in a UAA codon. | Plasmid NJM89; Strain NJM1249 | N/A | N/A | N/A | N/A | N/A |
| Genetic reagent (*E. coli*) | RK24 | 10.1126/science.1205822; 10.1016/j.cels.2016.06.009 | *pRK24*; Strain NJM699 | Conjugative RK2 plasmid (10.1006/jmbi.1994.1404), but lacks functional AmpR gene. | Strain NJM699 | N/A | N/A | N/A | N/A | N/A |
| Genetic reagent (*E. coli*) | F | Yale University Coli Genetic Stock Center (CGSC), Strain #4401 | *pF*; Strain EMG2; Strain CGSC#4401; Strain NJM426; Strain NJM473 | Conjugative F plasmid, as described by PMID: 4568763. Obtained from the Yale CGSC. | Strain NJM426; Strain NJM473 | N/A | N/A | N/A | N/A | N/A |
| Genetic reagent (*E. coli*) | pZE21_UAG-GFP | this paper | pZEtR-eGFP-cHis-TAG-v02; Plasmid NJM88; Strain NJM1242 | pZE21 plasmid with pLtetO promoter driving inducible expression of eGFP with a C-terminal 6-His tag and terminating in UAG codon. Inducible with anhydro-tetracycline. | Plasmid NJM88; Strain NJM1242 | N/A | N/A | N/A | N/A | N/A |

*Continued on next page*

*Continued*

| Reagent type (species) or resource | Designation | Source or reference | Identifiers | Additional information | Isaacs Lab Reference # | Full genotype of strains | # UAG Codons | RF1 status | Ribosomal rescue gene knockout | ssrA tag status |
|---|---|---|---|---|---|---|---|---|---|---|
| Genetic reagent (*E. coli*) | pZE21_UAA-GFP | this paper | pZEtR-eGFP-cHis-TAA-v02; Plasmid NJM89; Strain NJM1249 | pZE21 plasmid with pLtetO promoter driving inducible expression of eGFP with a C-terminal 6-His tag and terminating in UAA codon. Inducible with anhydro-tetracycline. | Plasmid NJM89; Strain NJM1249 | N/A | N/A | N/A | N/A | N/A |
| Genetic reagent (*Enterobacteria phage λ*) | λ.CI857 | Coli Genetic Stock Center (CGSC), Yale University (contact John Wertz directly) | λ.CI857; λ phage; Phage NJM102 | Phage λ with temperature-sensitive CI repressor gene; when incubated at 37° C, phage becomes obligate lytic | Phage NJM102 | N/A | N/A | N/A | N/A | N/A |
| Cell line (*E. coli*) | GRO.DD | this paper | C31GIB.tmRNA-DD; Strain #987 | MG1655-derived strain with all 321 UAG codons mutated to UAA, deletion of RF1, and tmRNA tag C-terminal amino acids mutated from AA to DD. Retains lambda red cassette for recombineering. Investigated in *Figure 2*. | Strain #987 | $\Delta mutS:zeo.\Delta(ybhB-bioAB):[\lambda cl857.\Delta(cro-ea59):tetR-bla].\Delta prfA.\Delta tolC.tmRNA_{DD}$ | 0 | +RF1 | n/a | DD |
| Cell line (*E. coli*) | GRO.DD.prfA+ | this paper | C31GIB.prfA+.tmRNA-DD; Strain #996 | MG1655-derived strain with all 321 UAG codons mutated to UAA, retains RF1, and tmRNA tag C-terminal amino acids mutated from AA to DD. Retains lambda red cassette for recombineering. Investigated in *Figure 2*. | Strain #996 | $\Delta mutS:zeo.\Delta(ybhB-bioAB):[\lambda cl857.\Delta(cro-ea59):tetR-bla].\Delta tolC.tmRNA_{DD}$ | 0 | ΔRF1 | n/a | DD |
| Cell line (*E. coli*) | ECNR2 | 10.1016/j.cels.2016.06.009 | ECNR2.ΔmutS:zeocin.ΔλRed; Strain #795 | MG1655-derived strain that contains 321 UAG codons and retains RF1. Investigated in *Figures 3* and *4*. | Strain #795 | $\Delta mutS:zeo$ | 321 | +RF1 | n/a | AA |
| Cell line (*E. coli*) | GRO.AA | 10.1016/j.cels.2016.06.009 | C31.final.ΔmutS:zeocin.ΔprfA.ΔλRed; Strain #796 | MG1655-derived strain with all 321 UAG codons mutated to UAA, deletion of RF1. Investigated in *Figures 3* and *4*. | Strain #796 | $\Delta mutS:zeo.\Delta prfA$ (GenBank ID: CP006698) | 0 | ΔRF1 | n/a | AA |

*Continued on next page*

*Continued*

| Reagent type (species) or resource | Designation | Source or reference | Identifiers | Additional information | Isaacs Lab Reference # | Full genotype of strains | # UAG Codons | RF1 status | Ribosomal rescue gene knockout | ssrA tag status |
|---|---|---|---|---|---|---|---|---|---|---|
| Cell line (*E. coli*) | GRO.AA.ΔarfB | this paper | C31GIB.arfB: tolCorf.ΔλRed; Strain #1230 | MG1655-derived strain with all 321 UAG codons mutated to UAA, deletion of RF1, and deletion of arfB. Investigated in *Figures 3* and *4*. | Strain #1230 | Δ*mutS:* zeo.Δ*prfA* .*arfB:tolC* | 0 | ΔRF1 | ΔssrA | AA |
| Cell line (*E. coli*) | GRO.AA.ΔssrA | this paper | C31GIB.ssrA :tolC.ΔλRed; Strain #1231 | MG1655-derived strain with all 321 UAG codons mutated to UAA, deletion of RF1, and deletion of ssrA. Investigated in *Figures 3* and *4*. | Strain #1231 | Δ*mutS:* zeo.Δ*prfA.* ssrA:tolC | 0 | ΔRF1 | ΔarfA | AA |
| Cell line (*E. coli*) | GRO.AA.ΔarfA | this paper | C31GIB.arfA :tolC.ΔλRed ; Strain #1232 | MG1655-derived strain with all 321 UAG codons mutated to UAA, deletion of RF1, and deletion of arfA. Investigated in *Figures 3* and *4*. | Strain #1232 | Δ*mutS:* zeo.Δ*prfA.* arfA:tolC | 0 | ΔRF1 | ΔarfB | AA |
| Cell line (*E. coli*) | GRO.AA .ΔssrA.ΔarfB | this paper | C31GIB.ΔarfB .ssrA:tolC.Δ λRed; Strain #1233 | MG1655-derived strain with all 321 UAG codons mutated to UAA, deletion of RF1, and deletion of ssrA and arfB. Investigated in *Figures 3* and *4*. | Strain #1233 | Δ*mutS:* zeo.Δ*prfA* .Δ*arfB.ssrA:tolC* | 0 | ΔRF1 | ΔssrA. ΔarfB | AA |
| Cell line (*E. coli*) | GRO.AA .ΔarfA. ΔarfB | this paper | C31GIB.Δarf B.arfA:tolC. ΔλRed; Strain #1234 | MG1655-derived strain with all 321 UAG codons mutated to UAA, deletion of RF1, and deletion of arfA and arfB. Investigated in *Figures 3* and *4*. | Strain #1234 | Δ*mutS:* zeo.Δ*prfA* .Δ*arfB.arfA* :*tolC* | 0 | ΔRF1 | ΔarfA. ΔarfB | AA |
| Antibody | mouse anti-GFP antibody | other | Invitrogen (Ref#: 332600, Lot#: 1513862A) | Invitrogen (Ref#: 332600, Lot#: 1513862A); (5.5 µL antibody in 3 mL Milk + TBST) | N/A | N/A | N/A | N/A | N/A | N/A |
| Antibody | goat anti-mouse antibody | other | AbCam (Ref#: ab7023, Lot#: GR157827-1) | AbCam (Ref#: ab7023, Lot#: GR157827-1); (2.2 µL antibody in 10 mL Milk + TBST) | N/A | N/A | N/A | N/A | N/A | N/A |
| Recombinant DNA reagent | ssrA:tolC | this paper; for use, see tolC positive /negative selection in 10.1038/nprot .2014.081 | dsDNA NJM111 | The E. coli native tolC gene used to delete ssrA gene via recombineering (10.1038/nprot. 2008.227). | dsDNA NJM111 | N/A | N/A | N/A | N/A | N/A |

*Continued on next page*

*Continued*

| Reagent type (species) or resource | Designation | Source or reference | Identifiers | Additional information | Isaacs Lab Reference # | Full genotype of strains | # UAG Codons | RF1 status | Ribosomal rescue gene knockout | ssrA tag status |
|---|---|---|---|---|---|---|---|---|---|---|
| Recombinant DNA reagent | arfA:tolC | this paper; for use, see tolC positive /negative selection in 10.1038/nprot .2014.081 | dsDNA NJM112 | The E. coli native tolC gene used to delete arfA gene via recombineering (10.1038/nprot. 2008.227). | dsDNA NJM112 | N/A | N/A | N/A | N/A | N/A |
| Recombinant DNA reagent | arfB:tolC | this paper; for use, see tolC positive /negative selection in 10.1038/nprot .2014.081 | dsDNA NJM113 | The E. coli native tolC gene used to delete arfB gene via recombineering (10.1038/nprot. 2008.227). | dsDNA NJM113 | N/A | N/A | N/A | N/A | N/A |
| Software, algorithm | Doubling time algorithm | 10.1126/ science.1241459 | Growth_ Analyze_ GK.m | Doubling time used in 10.1126 /science.1241459, written by Gleb Kuznetsov in the lab of Dr. George Church. | N/A | N/A | N/A | N/A | N/A | N/A |
| Software, algorithm | MaxQuant v1.5.1.2 | other | N/A | Commercial software for mass spectrometry analysis. | N/A | N/A | N/A | N/A | N/A | N/A |
| Software, algorithm | Graphpad Prism 7 | other | N/A | Commercial software for statistical analysis and graphing, provided through Yale University. | N/A | N/A | N/A | N/A | N/A | N/A |

## Strains and media

All bacteria used in this study are derived from *E. coli* ECNR2, which is in turn derived from *E. coli* MG1655 (GenBank ID: U00096) in which *mutS* is replaced by a zeocin resistance cassette (*Wang et al., 2009*; *Lajoie et al., 2013b*). Additionally, the native *bioAB* genes found in MG1655 are replaced by the lambda red cassette in ECNR2. This strain is designated ECNR2.AA (see *Table 1* for full genotype). For experiments expressing UAG-GFP and UAA-GFP for mass spectrometry, strains with all 321 UAG codons changed to UAA (designated 'GRO' strains) were used to control for potential differences in protein expression arising from these mutations (GenBank ID for GRO. AA: CP006698). For all other experiments, control strains labeled wild-type (WT) are MG1655 derivatives retaining all 321 UAG codons. All deletions of *ssrA*, *arfA*, and *arfB* were generated with a tolC resistance cassette via recombineering (*Sharan et al., 2009*). Modification of the *ssrA* tag from AANDENYALAA to AANDENYALDD (AA->DD) to increase stability of tagged proteins was performed with MAGE as described previously (*Gallagher et al., 2014*; *Wang et al., 2009*). All modifications to strains made in this study were validated through Sanger sequencing (GeneWiz; South Plainfield, NJ).

We performed all protein expression assays and conjugation assays in LB Lennox at pH 7.5. We performed all phage assays in Tryptone-KCl (TK) media as described previously (*Jaschke et al., 2012*; *Ma and Isaacs, 2016*; *Valentine et al., 2002*).

## Phages and plasmids

For viral relative titers, we used phage λ cI857 obtained from Dr. John Wertz at the Yale Coli Genetic Stock Center (CGSC) because it is obligately lytic at 37°C, preventing possible confounding

factors from lysogeny. We used the conjugative plasmid RK2 described in *Isaacs et al. (2011)*, which is a derivative of the RK2 plasmid described in *Pansegrau et al. (1994)* carrying bla$^R$ instead of kan$^R$. The complete nucleotide sequence for the plasmid is available in NCBI database, Accession L27758.1 and GI 508311. We obtained the F plasmid from the Yale CGSC (NCBI Accession AP001918.1, GI: 8918823) and added Kan$^R$ from plasmid pZE21 for antibiotic selection.

To create the UAG-GFP and UAA-GFP constructs for protein expression, we cloned an eGFP construct with a C-terminal 6xHis tag downstream of pLtetO into a modified pZE21 vector with kanamycin resistance (kan$^R$)carrying a copy of the tet repressor gene (tetR) to prevent leaked gene expression. We then modified the stop codon of the eGFP construct to end in either a UAG or UAA stop codon.

## Protein expression and purification

To obtain GFP for analysis via mass spectrometry, we transformed UAG-GFP and UAA-GFP constructs into wild-type and GRO strains carrying the AA->DD modification in the ssrA tag to prolong the half-life of tagged peptides. Experiments in the absence of the AA->DD modification yielded no peptides with ssrA degradation tags (data not shown). We then grew 50 mL cultures of each strain at 33°C in LB Lennox with 30 µg/mL kanamycin to an $OD_{600}$ of 1.0 and induced protein expression with the addition of 30 ng/uL anhydrotetracycline (aTC). After incubation overnight, we pelleted cells and resuspended them in sterile phosphate buffer solution, then lysed cells via sonication. Cell debris was then pelleted by centrifugation and GFP purified from supernatant via a nickel resin affinity column. To concentrate protein and exchange buffer for subsequent trypsin digest, we then concentrated GFP via Millipore Amicon spin columns.

For whole western blots on whole cell lysates, we transformed UAG-GFP and UAA-GFP constructs into wild-type, GRO, and GRO strains with deletions of the ribosomal rescue systems. We then grew 5 mL cultures of each strain at 33°C in LB Lennox with kanamycin overnight, then diluted all cultures $OD_{600}$ of 0.15 in fresh media containing 30 µg/mL kanamycin and 30 ng/uL aTC for 20 hr. To quantify protein expression and compare across strains, we normalized the $OD_{600}$ of all cultures to 2.5 and pelleted 1 mL of this culture, which we placed in the −80C for 2 hr. We then re-suspended cell pellets in lysis buffer described previously (*Aerni et al., 2015*), incubated for 10 min on ice, centrifuged lysate, and ran 1:10 dilutions of resulting supernatant on gels for western blot analysis. Overnight starter cultures were diluted to an $OD_{600}$ of 0.15 into three separate culture tubes, and cells within each tube were induced in parallel for GFP expression. GFP was purified from each of these cultures in parallel.

## Mass spectrometry and proteomic analysis

Trypsin digest, sample preparation for mass spectrometry, and liquid chromatography elution gradients were performed as described previously (*Aerni et al., 2015*). Desalted peptides were injected onto a 75 µm ID PicoFrit column (New Objective) packed to 50 cm in length with 1.9 µm ReproSil-Pur 120 Å C18-AQ (Dr. Maisch). Samples were eluted over a 90 min gradient using an EASY-nLC 1000 UPLC (Thermo) paired with a Q Exactive Plus (Thermo), using the following parameters: (MS1) 70,000 resolution, $3 \times 10^6$ AGC target, 300–1700 m/z scan range; (MS2) 17,500 resolution, $1 \times 10^6$ AGC target, top 10 mode, 1.6 m/z isolation window, 27 normalized collision energy, 90 s dynamic exclusion, unassigned and +1 charge exclusion. Peptide identification from collected spectra was performed using MaxQuant v1.5.1.2 (*Cox and Mann, 2008*). Samples were searched using custom databases representing potential translational outcomes in response to the UAG codon within the GFP reporter construct (*Supplementary file 3* and *4*), as well as the *E. coli* proteome (EcoCyc K-12 MG1655 v17). The searches considered carbamidomethyl (Cys) as a fixed modification and the following variable modifications: acetyl (N-terminal), oxidation (Met), deamidation (Asn, Gln), and phosphorylation (Ser/Thr/Tyr). Discovered peptides had a minimum length of five amino acids and could contain up to three trypsin miscleavage events. A 1% false discovery rate was used. The mass spectrometry proteomics data and the custom search databases have been deposited to the ProteomeX-change Consortium (http://proteomecentral.proteomexchange.org) via the PRIDE partner repository (*Vizcaíno et al., 2014*) with the dataset identifier PXD009643. Mass spectrometry spectra were manually validated by identifying all spectra with an MS/MS score over 15 and verifying the presence sufficient b- and/or y-ion series.

## Western blot experiments and analysis

Western blots were run as described previously using SDS-PAGE gels (*Pirman et al., 2015*). We ran GFP-6xHis standards of known amount (1, 10, 50, and 100 ng) alongside experimental samples and used these standards to generate linear-range calibration curves to quantify protein abundance in experimental samples (*Figure 3—figure supplement 1*). Because the antibody signal appeared sublinear in the 0–10 ng regime when we performed linear regression using all standards, we generated separate linear fits using the 1–10 ng standards and the 10–100 ng standards. We then determined experimental sample concentrations using these linear approximations. 20 of the 24 experimental samples quantified fell within or slightly above the 10–100 ng range (with the highest-intensity sample quantified as 136 ng), and 3 of the 24 samples fell within the 1–10 ng range. The one remaining sample, which had a weaker intensity than that of the 1 ng standard, was quantified through a linear approximation between the intensity of the 1 ng sample and of a blank lane with an assumed intensity of zero.

We expressed GFP-6xHis as described above, normalized cell cultures to an $OD_{600}$ of 2.5, and lysed cells using BugBuster protein extraction reagent (Merck, Darmstadt, Germany). We then ran 10 µl of 1/150 diluted lysate per lane of the SDS-PAGE gel. We obtained primary mouse anti-GFP antibody from Invitrogen (Ref#: 332600, Lot#: 1513862A; RRID:AB_2234927) and goat anti-mouse antibody from AbCam (Ref#: ab7023, Lot#: GR157827-1; RRID:AB_955413). Western blots were developed using Bio-Rad Clarity Western ECL Blotting Substrate and Imaged on a GE Amersham Imager 600. We performed quantification of western blot bands as described previously (*Pirman et al., 2015*). We repeated three western blots in parallel for each strain induced in separate culture tubes (i.e. biological triplicates, see Protein expression and purification).

## Viral relative titers

To quantify relative titers, we mixed 100-fold dilutions of phage with 300 µL of mid-log ($OD_{600}$ = 0.5) cells in 3 mL of TK soft agar and poured onto TK solid agar plates. Starter cultures of cells were diluted to an $OD_{600}$ of 0.5 into three separate culture tubes, and cells within each tube were infected with phage lambda in parallel (i.e. biological triplicate). Each tube was plated on a separate TK solid agar plate. We incubated plates overnight at 37°C, and counted plaques the next day.

## Quantifying conjugation

We used conjugation conditions described previously (*Ma and Isaacs, 2016*; *Ma et al., 2014*). Briefly, we grew cultures of donor and recipient cells to late log in antibiotics selecting for plasmid or recipient and then rinsed and re-suspended in media to remove antibiotics. After concentrating cells to an $OD_{600}$ of 20, we mixed donors and recipients in 1:1 ratio and spotted onto pre-warmed LB Lennox agar plates in 2 × 20 uL and 6 × 10 uL pattern. For F, we incubated plates at 37°C for 2 hr, then rinsed cells off plate, diluted serially 10-fold, and plated serial dilutions on plates containing antibiotic selecting for conjugants and incubated overnight at 37°C. For RK2, we incubated plates at 37°C for 1 hr, then plated on agar plates selecting for the recipient. To quantify the rate of transfer, we then picked 86 colonies from plates selecting for the recipient strain and patched them onto plates selecting for both recipient and conjugative plasmid, incubated plates overnight at 37°C, and counted the number of patched colonies that grew. After the conjugation, colonies were plated three times to generate technical triplicates.

## Statistical and data analysis

We performed all t-tests and one-way ANOVA tests for statistical significance in GraphPad Prism 7. We calculated doubling times and maximum $OD_{600}$ values from growth curve data using MATLAB (Newton, MA) code that we generated (*Source code 1*).

## Experimental replicates

We used the definitions for biological and technical replicates outlined in *Blainey et al., 2014*. Biological replicates consist of parallel measurements of different biological samples subjected to the same experiment, and technical replicates are parallel measurements of a single biological sample subjected to experimentation. Data represented in (*Figures 3*, *4B and D*) are biological replicates;

data represented in (*Figure 4A and C*) are technical replicates. Data for all 96-well plate assays (*Figures 3A, B* and *4B*) were obtained as biological replicates: One well of each sample was grown overnight as a starter culture in a 96-well plate. Starter cultures were then inoculated into three separate wells in a separate 96-well plate.

## Acknowledgements

We thank the members of the Isaacs and Rinehart labs for helpful feedback on this study. We also thank Paul Turner for valuable discussions and experimental advice. The authors gratefully acknowledge support from DARPA (N66001-12-C-4211, HR0011-15-C-0091 to FJI), DOE (152339.5055249.100 to FJI), NIH (R01GM117230, R01GM125951 to FJI and JR, T32GM007499 and T32GM007223 to NJM), and NSF (DGE-1122492 to KWB). The authors also thank the Gruber Foundation (NJM), the Arnold and Mabel Beckman Foundation (FJI and CFH), and DuPont Inc. (FJI) for funding.

## Additional information

### Funding

| Funder | Grant reference number | Author |
| --- | --- | --- |
| Defense Advanced Research Projects Agency | N66001-12-C-4211 | Farren Isaacs |
| U.S. Department of Energy | 152339.5055249.100 | Farren Isaacs |
| National Institutes of Health | R01GM117230 | Jesse Rinehart<br>Farren Isaacs |
| National Institutes of Health | R01GM125951 | Farren J Isaacs<br>Jesse Rinehart |
| National Science Foundation | Graduate Research Fellowship DGE-1122492 | Karl W Barber |
| Gruber Foundation | Graduate Research Fellowship | Natalie Jing Ma |
| Arnold and Mabel Beckman Foundation | Young Investigator Award | Farren Isaacs |
| DuPont | Young Professor Award | Farren Isaacs |
| National Institutes of Health | Graduate Training Grants T32GM007499, T32GM007223 | Natalie Jing Ma |
| Defense Advanced Research Projects Agency | HR0011-15-C-0091 | Farren Isaacs |
| Arnold and Mabel Beckman Foundation | Beckman Scholar Award | Colin F Hemez |

The funders had no role in study design, data collection and interpretation, or the decision to submit the work for publication.

### Author contributions

Natalie Jing Ma, Conceptualization, Formal analysis, Validation, Investigation, Visualization, Methodology, Writing—original draft, Writing—review and editing; Colin F Hemez, Investigation, Visualization, Writing—original draft, Writing—review and editing; Karl W Barber, Resources, Investigation, Visualization, Writing—review and editing; Jesse Rinehart, Resources, Supervision, Funding acquisition, Validation, Methodology; Farren J Isaacs, Conceptualization, Supervision, Funding acquisition, Methodology, Writing—original draft, Writing—review and editing

**Author ORCIDs**

Natalie Jing Ma  https://orcid.org/0000-0001-7452-9482
Colin F Hemez  http://orcid.org/0000-0003-3445-7706
Karl W Barber  http://orcid.org/0000-0003-0672-8409
Farren J Isaacs  https://orcid.org/0000-0001-8615-8236

**Decision letter and Author response**

Decision letter https://doi.org/10.7554/eLife.34878.033
Author response https://doi.org/10.7554/eLife.34878.034

# Additional files

**Supplementary files**

• Source code 1. MATLAB script used to analyze growth curve data from 96-well plate assays (*Figure 3A,B* and *4B*).
DOI: https://doi.org/10.7554/eLife.34878.024

• Supplementary file 1. Gene nucleotide sequences, processed mass spectrometry data, and numerical values used to generate *Figures 2–4*.
DOI: https://doi.org/10.7554/eLife.34878.025

• Supplementary file 2. Spectra for all 47 manually verified peptides detected through mass spectrometry from GRO.AA expressing the UAG-GFP plasmid.
DOI: https://doi.org/10.7554/eLife.34878.026

• Supplementary file 3. Library of peptides generated for the detection of frameshifting, near-cognate suppression, and ribosomal rescue events from mass spectrometry data.
DOI: https://doi.org/10.7554/eLife.34878.027

• Supplementary file 4. Library of peptides generated for the detection of loss of translational fidelity from mass spectrometry data.
DOI: https://doi.org/10.7554/eLife.34878.028

• Transparent reporting form
DOI: https://doi.org/10.7554/eLife.34878.029

**Data availability**

Sequences of strains used have been previously published with the appropriate citations. Modifications (e.g., gene deletions) to those strains are described in full in the Tables, Key Resource Guide, methods and supplementary material. The mass spectrometry proteomics data have been deposited to the ProteomeXchange Consortium via the PRIDE partner repository with the dataset identifier PXD009643 (http://proteomecentral.proteomexchange.org) via the PRIDE partner repository (Vizcaíno et al, 2014).

The following dataset was generated:

| Author(s) | Year | Dataset title | Dataset URL | Database, license, and accessibility information |
| --- | --- | --- | --- | --- |
| Jing Ma N, Hemez CF, Barber KW, Rinehart J, Isaacs F | 2018 | Mass spectrometry proteomics data from "Organisms with alternative genetic codes resolve unassigned codons via mistranslation and ribosomal rescue" | http://proteomecentral.proteomexchange.org/cgi/GetDataset?ID=PXD009643 | Publicly available at ProteomeXchange (accession no: PXD009643) |

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
