## [Decision Letter]

Thank you for submitting your article "Organisms with alternative genetic codes resolve unassigned codons via mistranslation and ribosomal rescue" for consideration by *eLife*. Your article has been reviewed by Gisela Storz as the Senior Editor, a Reviewing Editor, and three reviewers. The following individuals involved in review of your submission have agreed to reveal their identity: Kenneth Keiler (Reviewer #1); Yitzhak Pilpel (Reviewer #2); Alexander Mankin (Reviewer #3).

The reviewers have discussed the reviews with one another and the Reviewing Editor has drafted this decision to help you prepare a revised submission.

In this manuscript, Ma et al. investigate how bacteria contend with unassigned codons by expressing genes with a UAG stop codon in a genetically recoded *E. coli* strain lacking UAG codons and RF1. This question is important because there are a number of examples in nature of codon reassignment, but it is not clear how organisms would make the transition from a canonical code if there is a large penalty for having an unassigned codon. It has also been suggested that reassignment might limit transfer of foreign DNA. The authors used mass spectrometry to identify the proteins produced from a reporter gene with a UAG codon and find that multiple mechanisms help cells deal with the unassigned codon, including suppression, frameshifting, and general mistranslation. The authors also tested the effect of inactivation of three ribosome rescue systems (tmRNA, ArfA and ArfB) on the reporter expression and on propagation of conjugative plasmids or phages that have genes terminating in UAG. These data suggested that tmRNA accounts for the resistance of their *E. coli* mutant strain to horizontal gene transfer.

This is an interesting study that expands our knowledge of how cells deal with difficult-to-translate codons. It is also useful for the future use of genetically recoded organisms.

Essential revisions:

1) The manuscript consistently refers to unassigned codons, implying that the work addresses this general issue. For example, in the Introduction: "Our work reveals mechanistic details into how cells rescue ribosomes stalled at unassigned codons". However, the effects observed when ribosome rescue pathways are deleted are almost certainly unique to unassigned stop codons. Trans-translation activity on ribosomes stalled at unassigned stop codons will remove the protein, but when *ssrA* is deleted, ArfA will release a complete polypeptide identical to what would be produced by RF1. If a sense codon were unassigned, neither trans-translation nor ArfA activity could produce active proteins. The authors should either provide an explanation for why the ribosome rescue pathways would have the same impact on unassigned codons within a gene, or they should clarify that their interpretation is restricted to unassigned stop codons.

2) For the experiment in Figure 2, the basis for some of the assignments are unclear. For example, how is it known that LEHHHHHHMVR results from a +19 skip instead of from loss of fidelity like LEHHHHHHYQR? The presence of two additional His residues in the His-tail of the GFP-His6 reporter may indeed indicate a -6 frameshift, as the authors propose, but may also indicate two consecutive -3 frameshifting events. Authors do not discuss the second possibility and suggest that they have detected "the furthest frameshift backward". Without strong evidence that they are indeed dealing with a -6 frameshift, instead of two -3 frameshifts, this seems to be an overstatement. Some of the peptides could also result from transcription errors rather than translation mistakes. A more quantitative summary of the types of peptides found in the mass spec, including the types of peptides found in the strain expressing the UAA-containing construct, would be very useful.

3) Were the experiments in Figure 3 and Figure 4 done with strains containing *ssrA*-DD (as in Figure 2) or wild-type *ssrA*? This is critical for the interpretation. All strains and plasmids need to be described at a level of detail that would allow others to reproduce the work.

4) Figure 3A is completely not clear and perhaps even misleading since it shows the effect of the different mutations on the *ratio* between the expressed and non-expressed constructs. It is hard to know whether the effects shown are due to effects of the mutations on the expressed or non-expressed construct. To resolve this issue, the authors should show the effects of the different mutations on the actual max OD_600_ and doubling time values of each strain separately (i.e. not just on the ratio).

5) The results shown for some of the strains (Figure 3) are surprising and not expected (for example, the large decrease in max OD_600_ ratio of the *ssrA arfB* double mutant – Figure 3A, the opposite trend in GFP values shown for the *ssrA arfB* compared to *arfB* and *ssrA* alone – Figure 3C etc.). There is no clear explanation to these peculiarities in the text. A more elaborate discussion in the context of epistatic interactions shown is needed, such a discussion could address the effects of double mutations compare to single ones. Likewise, the authors should try to comment on the observation that knockout of *arfA* increases GFP production (Figure 3C), whereas ArfA is expected to facilitate termination of the GFP translation at the UAG codon.

6) In Figure 4B, the authors interpreted the change in doubling time as an indication for the ability of RK2 plasmid to replicate – some explanation to this should be added to text.

[Editors' note: further revisions were requested prior to acceptance, as described below.]

Thank you for resubmitting your work entitled "Organisms with alternative genetic codes resolve unassigned codons via mistranslation and ribosomal rescue" for further consideration at *eLife*. Your revised article has been favorably evaluated by Gisela Storz (Senior Editor) and a Reviewing Editor.

The manuscript has been improved but there are some remaining issues that need to be addressed before acceptance, as outlined below:

1) For the experiments in Figure 3 and Figure 4, explain whether the replicates are biological replicates or technical replicates and at what step the replication occurred. From the source data, it appears that all of the growth data is from a single 96 well plate. Were replicate wells in the plate inoculated from independent overnight cultures or from the same culture? This distinction is important to determine if the small changes observed are likely to be physiologically relevant. It is confusing that there are 3 plates of data in the source file, but data in the figure all seems to come from a single plate. Likewise, for the conjugation experiments, do the data come from one mating that was plated three times, from three matings made with the same cultures, or from three matings performed with independently grown cultures?

2) Figure 3C shows negative protein concentration, and the explanation is that the band was quantified, and the value was plugged into a formula from a standard curve, giving a negative result. Because it is physically impossible to have negative protein concentration and there is quite clearly a band on the blot, the explanation provided suggests that the standard curve was not accurate. From the source data, it appears that the standard curves are derived from two points at 1 ng and 100 ng fit to a line. Unfortunately, almost all of the samples are outside the 1-100 ng range. It appears the negative value is the result of a large negative number from blot 3 which also seems to have an anomalously low value for 100 ng in the standard curve. The differences in protein production from the different samples are clear, but the quantification is clearly inaccurate. This problem could potentially be addressed by running another replicate of the experiment or reporting the data normalized to one of the samples such as GRO.AA [pUAG-GFP]. We will not publish a negative concentration.

Text clarifications:a) In subsection “Suppression, ribosomal frameshifting, and *ssrA* tagging occur at unassigned codons”: Spontaneous termination of translation could refer to untemplated termination by RF2 or to spontaneous (nonenzymatic) hydrolysis of the peptidyl-tRNA.

b) In subsection “*ssrA* and *arfB* mediate degradation of proteins containing unassigned UAG codons”: Fitness is best used in relation to competitive growth experiments because it is possible for a strain to grow to a lower OD_600_ in monoculture but outcompete other strains and therefore have higher fitness. It would be clearer here to refer to growth rate or OD_600_ instead of fitness.

c) In subsection “*ssrA* and *arfB* mediate degradation of proteins containing unassigned UAG codons”: What is the knockout of *arfB* being compared to here? In Figure 3A, the induced/uninduced ratio for the *arfB* strain looks very similar to that for the isogenic *arfA^+^* strain, which would seem to suggest that ArfB does not play a role.

d) The sentence in the last paragraph of subsection “*ssrA* and *arfB* mediate degradation of proteins containing unassigned UAG codons” is unclear: "Interestingly, a single knockout of arfB significantly reduced production of protein from UAG-GFP to low levels similar to those mapped to quantified GFP standards."

e) In subsection “*ssrA* and *arfB* mediate degradation of proteins containing unassigned UAG codons”, the comparison for "fully restore protein expression from UAG-ending transcripts" appears to be GRO.AA[pUAA-GFP] instead of ECNR2[pUAG-GFP], but "restore" suggests it should be the latter. The comparison and the meaning behind which strain is used for the comparison should be clarified.

f) In subsection “Deletion of *ssrA* restores conjugative plasmid propagation and viral infection in the GRO”, it is not clear what comparison was used for the 2.4-fold increase in doubling time. The graph in Figure 4B shows a 38% increase for the *arfA* strain versus 28% for the isogenic wild type.

g) Discussion section: What is the evidence supporting the demonstration of ribosome stalling? It seems that ribosome stalling was assumed based on the addition of the SsrA tag. Experiments such as ribosome profiling could demonstrate ribosome stalling, but these were not done. I think stalling is part of the model here but has not been demonstrated.

h) Discussion section: Similar to (g) above, the regulatory relationship between tmRNA and ArfA was used to explain the data, so it would be circular reasoning to then use this explanation to validate the regulatory relationship.

i) Discussion section: "extensive" suggests there is a large amount of frameshifting, but the frameshifting events cannot be quantified using the techniques in this work. Perhaps something like "a wide variety of frameshifting events" would be more accurate.

j) Figure 1: The cartoon shows the SsrA-tagged protein going into the protease N terminus first, but all the proteases recognize the tag and start at the C terminus (this is not critical). In the legend, the word "hypothesized" should be removed from the last sentence – the lack of modification has been observed.

k) Figure 3 legend: For panels A and B, the legend does not match the labels. A is doubling time and B is max OD.

---

## [Author Response]

In this manuscript, Ma et al. investigate how bacteria contend with unassigned codons by expressing genes with a UAG stop codon in a genetically recoded E. coli strain lacking UAG codons and RF1. This question is important because there are a number of examples in nature of codon reassignment, but it is not clear how organisms would make the transition from a canonical code if there is a large penalty for having an unassigned codon. It has also been suggested that reassignment might limit transfer of foreign DNA. The authors used mass spectrometry to identify the proteins produced from a reporter gene with a UAG codon and find that multiple mechanisms help cells deal with the unassigned codon, including suppression, frameshifting, and general mistranslation. The authors also tested the effect of inactivation of three ribosome rescue systems (tmRNA, ArfA and ArfB) on the reporter expression and on propagation of conjugative plasmids or phages that have genes terminating in UAG. These data suggested that tmRNA accounts for the resistance of their E. coli mutant strain to horizontal gene transfer.This is an interesting study that expands our knowledge of how cells deal with difficult-to-translate codons. It is also useful for the future use of genetically recoded organisms.

We thank the reviewers for recognizing the significance of our work and relevance to both the translation of rare codons found in nature and to the translation of codons engineered into genomically recoded organisms.

Essential revisions:1) The manuscript consistently refers to unassigned codons, implying that the work addresses this general issue. For example, in the Introduction: "Our work reveals mechanistic details into how cells rescue ribosomes stalled at unassigned codons". However, the effects observed when ribosome rescue pathways are deleted are almost certainly unique to unassigned stop codons. Trans-translation activity on ribosomes stalled at unassigned stop codons will remove the protein, but when ssrA is deleted, ArfA will release a complete polypeptide identical to what would be produced by RF1. If a sense codon were unassigned, neither trans-translation nor ArfA activity could produce active proteins. The authors should either provide an explanation for why the ribosome rescue pathways would have the same impact on unassigned codons within a gene, or they should clarify that their interpretation is restricted to unassigned stop codons.

It is true that our work is limited to unassigned stop codons and we agree that ribosomal rescue through trans-translation or ArfA at an unassigned sense codon would likely produce a truncated, nonfunctional protein and think this greatly supports our argument made in the Discussion section that further modifying the genetic code in engineered organisms could lead to even greater barriers to horizontal gene transfer. However, we also recognize prior literature that reports that these ribosomal rescue mechanisms would not be restricted to unassigned stop codons. As seen previously, these ribosomal rescue mechanisms are active at rare sense codons (Hayes, Bose and Sauer, 2002) and during periods of amino acid starvation (George et al., 2016; Li et al., 2007; Roche and Sauer, 1999), as we note in the Discussion section, suggesting they would also rescue stalled ribosomes at unassigned sense codons. In this manuscript, we empirically examine this in the context of an unassigned stop codon for two reasons: (1) when *ssrA* is deleted, fully-functional protein is made that is detectable both by western blot for GFP and phenotypic rescue for conjugative plasmids and viruses, which simplifies study of the molecular mechanism and (2) our model remains the only organism to date engineered to have an unassigned codon. Given that recoding of sense codons is actively underway (Napolitano et al., 2016; Ostrov et al., 2016; Lau et al., 2017), such GROs with unassigned sense codons would offer opportunities to conduct a similar study on sense codons to directly answer such questions.

2) For the experiment in Figure 2, the basis for some of the assignments are unclear. For example, how is it known that LEHHHHHHMVR results from a +19 skip instead of from loss of fidelity like LEHHHHHHYQR? The presence of two additional His residues in the His-tail of the GFP-His6 reporter may indeed indicate a -6 frameshift, as the authors propose, but may also indicate two consecutive -3 frameshifting events. Authors do not discuss the second possibility and suggest that they have detected "the furthest frameshift backward". Without strong evidence that they are indeed dealing with a -6 frameshift, instead of two -3 frameshifts, this seems to be an overstatement. Some of the peptides could also result from transcription errors rather than translation mistakes. A more quantitative summary of the types of peptides found in the mass spec, including the types of peptides found in the strain expressing the UAA-containing construct, would be very useful.

It would be difficult to determine using mass spectrometry whether LEHHHHHHMVR results from a + 19 skip or a loss of fidelity, but given that the skips of up to +50 bases have been validated in vivo (Huang et al., 1988) and loss of fidelity has only been observed in vitro prior to this manuscript, we hypothesize that this peptide arises from skipping. In addition, a key difference between LEHHHHHHMVR and the 5 tripeptides described in subsection “Suppression, ribosomal frameshifting, and *ssrA* tagging occur at unassigned codons” is that LEHHHHHHMVR could be genetically encoded with a +19 skip, while the other peptides could not be encoded unless multiple mistranslation events occurred. With regard to LEHHHHHHHH resulting from a -6 frameshift or two -3 frameshifts, it would be impossible to distinguish between these two scenarios using mass spectrometry data. As we have not found anything in the literature to denote which scenario is more likely, we have modified both the Abstract and text in subsection “Suppression, ribosomal frameshifting, and *ssrA* tagging occur at unassigned codons”, to read: “We detected ribosomal frameshifting of up to -3 (LEHHHHHHH) and +19 nucleotides (LEHHHHHHMVR), as determined by presence of fragments from all three reading frames appended immediately following the C-terminal peptide of LEHHHHHH. Additionally, the LEHHHHHHHH peptide may indicate a -6 frameshift, although it is not possible to determine whether this peptide arises from a single -6 frameshift or two -3 frameshifts between histidine incorporation.”

Although some of the tripeptides we identified could be the result of transcription mistakes, the number of transcriptional mistakes required to produce the degenerate tripeptides we identified would be between 2 and 6 transcriptional errors in a 30-nucleotide span. Given that the transcriptional error rate is 1 in 10,000 bases (Blank et al., 1986; Rosenberger and Hilton, 1983) and our strains have no mutations that would lead to greater error rates in transcription, we find this hypothesis to be less likely than loss of translational fidelity. We have revised the text to clarify this in subsection “Suppression, ribosomal frameshifting, and *ssrA* tagging occur at unassigned codons”, which reads: “it is unlikely these fragments arose from routine errors in mRNA transcription because this would require ≥2 transcriptional errors in a 30-nucleotide span. The transcription error rate in *E. coli* is estimated to be ~1 in 10,000 bases (Blank et al., 1986; Rosenberger and Hilton, 1983) and our strains have no known mutations that would lead to greater error rates in transcription.”

A summary of the C-terminal peptides identified from our mass spectrometry data is available on Supplementary file 1 and Supplementary file 2, and this has been modified to include the source strain for each peptide identified. As noted in the manuscript in subsection “Suppression, ribosomal frameshifting, and *ssrA* tagging occur at unassigned codons”, the only detectable GFP C-terminal peptide from UAA-ending constructs (and from strains containing RF1) were LEHHHHHH and LEHHHHHHAANDENYALDD. Mass spectrometry datasets that include ion intensity scores are available in Supplementary file 1 and Supplementary file 2. We agree that the quantification of the relative abundances of the various observed translational products would be very useful. Unfortunately, the amino acid sequence of each unique tryptic peptide confers different ionization properties, meaning that some peptides are much more amenable to observation by mass spectrometry than others. By the same token, the intensity of the peptides observed by mass spectrometry cannot be used to directly quantify difference in abundance between different peptides and mistranslated proteins. We therefore view our experiments as a “scouting” technique to identify the possible translational outcomes in response to the unassigned UAG codon, but further experimentation using a targeted collection of isotopically labeled peptides would be necessary for absolute peptide quantification, which is beyond the scope of the current work. However, even rigorously quantitative measures pose challenges because protein turnover rate is difficult to assess, and quickly-degraded proteins such as those with the tmRNA tag may be undervalued by simply comparing peptide abundances. A detailed description of mass spectrometry and proteomic analysis was added to the Materials and methods section of the manuscript.

3) Were the experiments in Figure 3 and Figure 4 done with strains containing ssrA-DD (as in Figure 2) or wild-type ssrA? This is critical for the interpretation. All strains and plasmids need to be described at a level of detail that would allow others to reproduce the work.

We thank the reviewers for their comment on the *ssrA* variants used in Figure 3 and Figure 4, and for pointing out the need to describe the strains and plasmids used in this study in greater detail. Strains used for the protein yield (Figure 3) and conjugation/infection (Figure 4) experiments contain wild-type *ssrA (ssrA*-AA). We added the following sentence to the manuscript to clarify this (subsection “*ssrA* and *arfB* mediate degradation of proteins containing unassigned UAG codons”): “Since mass spectrometry data indicated that a combination of mechanisms could resolve stalled translation at the unassigned UAG codon, we constructed targeted deletions of the ribosomal rescue systems (*ssrA, arfA*, and *arfB*) in strains with wild-type *ssrA* sequence (GRO.AA) to determine whether protein production from UAG-ending transcripts in ΔRF1 cells could be restored to levels seen in +RF1 cells.”

We have also compiled a new table of strains (Table 1) and a list of plasmids (see Key Resources Table) used in this study. The strain table includes information on the status of the *ssrA* tag, as well as the figure attribution for each strain.

*4) Figure 3A is completely not clear and perhaps even misleading since it shows the effect of the different mutations on the* ratio *between the expressed and non-expressed constructs. It is hard to know whether the effects shown are due to effects of the mutations on the expressed or non-expressed construct. To resolve this issue, the authors should show the effects of the different mutations on the actual max OD_600_ and doubling time values of each strain separately (i.e. not just on the ratio).*

We thank the reviewers for this comment and agree with their assessment. We have thus modified Figure 3 to consist of bar graphs demonstrating the doubling times with and without GFP expression (Figure 3A) and max OD_600_with and without GFP expression (Figure 3B). The quantitative values that comprise these figures are also available on Supplementary file 3.

5) The results shown for some of the strains (Figure 3) are surprising and not expected (for example, the large decrease in max OD600 ratio of the ssrA arfB double mutant – Figure 3A, the opposite trend in GFP values shown for the ssrA arfB compared to arfB and ssrA alone – Figure 3C etc.). There is no clear explanation to these peculiarities in the text. A more elaborate discussion in the context of epistatic interactions shown is needed, such a discussion could address the effects of double mutations compare to single ones. Likewise, the authors should try to comment on the observation that knockout of arfA increases GFP production (Figure 3C), whereas ArfA is expected to facilitate termination of the GFP translation at the UAG codon.

We thank the reviewers for raising these points and would hypothesize that, given limited cellular resources, cells are capable of expressing high levels of protein or achieving high fitness (via low doubling times or high max OD_600_) but not both simultaneously. In *ssrA* knockout strains, GFP expression increase at the expense of cellular fitness because GFP peptides stalled on ribosomes are freed by ArfA or ArfB and not tagged for degradation. We have clarified this point in the manuscript in subsection “*ssrA* and *arfB* mediate degradation of proteins containing unassigned UAG codons”: “These *ssrA* knockout strains likely demonstrate increased GFP expression and reduced fitness (Figure 3A and 3B) because translation of GFP transcripts sequester cellular resources at the expense of cellular replication, producing GFP peptides that are freed from stalled ribosomes via ArfA or ArfB without addition of a degradation tag.”

We have also addressed the unusual results in the *arfB* knockout strain by stating in subsection “*ssrA* and *arfB* mediate degradation of proteins containing unassigned UAG codons”: “Interestingly, a single knockout of *arfB* significantly reduced production of protein from UAG-GFP to an undetectable level when calibrated to quantified GFP standards (p = 0.0311). These ArfB deletion data, together with the fitness reduction observed in the GRO, suggest that ArfB is constitutively expressed and relieving low levels of ribosomal stalling in *E. coli*.”

Since the increased GFP expression observed in the *arfA* knockout strain is not statistically significant when compared to the GRO strain expressing the same UAG-GFP construct, we do not think emphasizing such nuanced differences is warranted and would require detailed follow-up in future work.

6) In Figure 4B, the authors interpreted the change in doubling time as an indication for the ability of RK2 plasmid to replicate – some explanation to this should be added to text.

We thank the reviewers for identifying the need to clearly articulate our interpretation of doubling time data. The basis for interpreting changes in doubling time as an indication of RK2’s ability to replicate stems from a prior study we conducted that demonstrated the UAG-ending *trfA* gene on RK2, which initiates plasmid replication, reduces cell growth and increases doubling time in the GRO unless its UAG stop codon is recoded to UAA when grown in media selecting for plasmid maintenance (Ma and Isaacs, 2016). To clarify this point, we changed the sentence in which we discuss these results (in subsection “Deletion of *ssrA* restores conjugative plasmid propagation and viral infection in the GRO”) to the following: “Previous research indicates that the UAG stop codon in the *trfA* gene on RK2 leads to impaired conjugation efficiency and replication in the GRO (Ma and Isaacs, 2016), likely because the trfA protein is required to initiate plasmid replication(Pansegrau et al., 1994). Phenotypically, this manifests as reduced efficiency of plasmid transfer in conjugation experiments and increased doubling times for RK2^+^ strains in media selecting for plasmid maintenance due to loss of plasmid and concomitant antibiotic resistance genes.”

[Editors' note: further revisions were requested prior to acceptance, as described below.]

The manuscript has been improved but there are some remaining issues that need to be addressed before acceptance, as outlined below:1) For the experiments in Figure 3 and Figure 4, explain whether the replicates are biological replicates or technical replicates and at what step the replication occurred. From the source data, it appears that all of the growth data is from a single 96 well plate. Were replicate wells in the plate inoculated from independent overnight cultures or from the same culture? This distinction is important to determine if the small changes observed are likely to be physiologically relevant. It is confusing that there are 3 plates of data in the source file, but data in the figure all seems to come from a single plate. Likewise, for the conjugation experiments, do the data come from one mating that was plated three times, from three matings made with the same cultures, or from three matings performed with independently grown cultures?

Thank you for pointing out the need to distinguish between biological and technical replicates in our experiments. We used the definitions for biological and technical replication outlined in Blainey et al., (2014): Biological replicates are parallel measurements of different biological samples subjected to the same experiment, and technical replicates are parallel measurements of a single biological sample subjected to experimentation. The data represented in all figures except for Figure 4A and 4C are the result of biological replicates. Figure 4A and 4C represent technical replicates (see below for more details).

The source data file for Figure 3A and 3B (Figure 3—source data 2) contains data from three identical 96-well plate experiments conducted using different 96-well plate reader models. Each experiment tested each sample in biological triplicate. Although data from the first two experiments are consistent with data from the third experiment, variability among plate reader machines precludes us from averaging the data obtained in the different experiments to draw conclusions of biological relevance. As such, we only represent the data obtained from “Plate 3” in Figure 3A and 3B. We modified the legend for Figure 3—source data 2 to clarify this: “Analysis of doubling times and maximum OD_600_’s of indicated strains. File contains doubling times and maximum OD_600_’s for three separate experiments conducted on different plate reader machines. Each experiment tested each sample in biological triplicate. Only the biological triplicate data from Plate 3 is represented in Figure 3A and 3B.”

To clarify that the data obtained from our conjugation experiments (represented in Figure 4A and 4C) are the result of technical replication, we have added the following sentence to the legend for Figure 4A and 4C: “Data are obtained from technical triplicates generated from a single biological sample.” We also note that these data are the consequence of technical replication in the legends for the appropriate source data files (Figure 4—source data 1 and Figure 4—source data 4).

We have written a new paragraph within our Materials and methods section to outline our definitions of biological and technical replication, and have clarified the nature of replication for each experiment within relevant paragraphs of the Materials and methods section. We updated our Transparent Reporting Form to include the above-mentioned definitions of biological and technical replication.

2) Figure 3C shows negative protein concentration, and the explanation is that the band was quantified, and the value was plugged into a formula from a standard curve, giving a negative result. Because it is physically impossible to have negative protein concentration and there is quite clearly a band on the blot, the explanation provided suggests that the standard curve was not accurate. From the source data, it appears that the standard curves are derived from two points at 1 ng and 100 ng fit to a line. Unfortunately, almost all of the samples are outside the 1-100 ng range. It appears the negative value is the result of a large negative number from blot 3 which also seems to have an anomalously low value for 100 ng in the standard curve. The differences in protein production from the different samples are clear, but the quantification is clearly inaccurate. This problem could potentially be addressed by running another replicate of the experiment or reporting the data normalized to one of the samples such as GRO.AA [pUAG-GFP]. We will not publish a negative concentration.

We agree with the reviewers that reporting a negative GFP value does not make sense, and we have amended our data analysis to provide better quantitation of the amount of GFP present in each sample. Images of the blots were provided in our initial submission as Figure 3—source data 3-5. The raw values for our reanalysis are provided in an updated version of Figure 3—source data 6.

For the western blots represented in Figure 3C, 1ng, 10ng, 50ng and 100ng GFP-6xHis standards were run on each blot next to the experimental samples. The majority of the samples fell within this range of GFP standards because the y-axis of Figure 3C, in µg/mL GFP, corresponds to a 6x correction factor based on the loading volume of lysate in each lane of the SDS-PAGE gels, to convert the ng of protein detected to µg/mL in the total lysate. Our previous data analysis, which yielded negative values for the GRO.AA*.∆arfB* samples and the 1ng standard, suggested that the antibody signal is sublinear in the 0-10ng regime. To address this problem, we revised the data analysis as follows: we created new standard curves using only the 10ng, 50ng and 100ng standards for each replicate western blot (R^2^ > 0.97 for all three blots), and fit data points with an intensity greater than that of the 10 ng standard to this curve. 20 of 24 samples fell within or slightly above this range, with the highest-intensity sample quantified as 136ng. We created a second set of standard curves using the 1ng and 10ng standards to quantify the protein abundances of three samples whose intensities fell within this range. The one remaining sample had a weaker intensity than that of the 1ng standard, and we quantified protein in this sample through a linear approximation between the 1ng sample intensity and a blank lane with an intensity of zero.

The results of this analysis are not substantively different than our previous analysis, but this method is clearly superior in calculating the GFP concentration from low-intensity samples and allows us to accurately report non-negative protein abundances. The standard curves from our reanalysis are provided in Figure 3—figure supplement 1.

We have revised the subsection “Western blot” to reflect these changes and to describe our analysis procedure in greater detail. The subsection now reads: “Western blots were run as described previously using SDS-PAGE gels (Pirman et al., 2015). […] We repeated three western blots in parallel for each strain induced in separate culture tubes (i.e., biological triplicates, see Protein expression and purification).”

Text clarifications:a) In subsection “Suppression, ribosomal frameshifting, and ssrA tagging occur at unassigned codons”: Spontaneous termination of translation could refer to untemplated termination by RF2 or to spontaneous (nonenzymatic) hydrolysis of the peptidyl-tRNA.

We thank the reviewers for pointing this out. The nonenzymatic hydrolysis of peptidyl-tRNA is posited to be a rare event (occurring about once per 100,000 codon decoding events), and we refer to this possibility in subsection “Suppression, ribosomal frameshifting, and tmRNA-mediated peptide *ssrA* tagging occur at unassigned codons”. However, we recognize that our methods do not distinguish between these possible events and have thus modified subsection “Suppression, ribosomal frameshifting, and tmRNA-mediated peptide *ssrA* tagging occur at unassigned codons” to read: “Given this, we hypothesize that these five peptides result from loss of translational fidelity after stalling at the UAG codon that may lead to (1) spontaneous termination of translation due to the untemplated action of RF2 following mistranslation or (2) ArfA- or ArfB-mediated release predicated on 3’ exonuclease degradation of the mRNA. The rare event of spontaneous hydrolysis of the peptide from the ribosome is also possible.”

b) In subsection “ssrA and arfB mediate degradation of proteins containing unassigned UAG codons”: Fitness is best used in relation to competitive growth experiments because it is possible for a strain to grow to a lower OD600 in monoculture but outcompete other strains and therefore have higher fitness. It would be clearer here to refer to growth rate or OD600 instead of fitness.

We thank the reviewers for making this distinction. We have replaced the term “fitness” in subsection “Suppression, ribosomal frameshifting, and tmRNA-mediated peptide *ssrA* tagging occur at unassigned codons” to “growth rate and cell density.”

c) In subsection “ssrA and arfB mediate degradation of proteins containing unassigned UAG codons”: What is the knockout of arfB being compared to here? In Figure 3A, the induced/uninduced ratio for the arfB strain looks very similar to that for the isogenic arfA^+^ strain, which would seem to suggest that ArfB does not play a role.

In the sentence that the reviewers mention, we are comparing GFP production from UAG-ending transcripts between a strain with *ssrA* knockout and a strain with both an *ssrA* and *arfB* knockout. We intend to point out that the knockout of both *ssrA* and *arfB* is needed to recover GFP production to levels seen in GRO cells that produce GFP from UAA-ending transcripts. We revised this sentence (and the sentence before it) to clarify the nature of our comparisons, and to make explicit the nature of the p-values we are reporting (all of which are calculated using GRO.AA [pUAG-GFP] as the null hypothesis). The paragraph beginning in subsection “*ssrA* and *arfB* mediate degradation of proteins containing unassigned UAG codons” now reads:

“We then investigated the impact of unassigned codons on protein production using western blot densitometry, and found that the GRO expressing UAG-GFP produced less than one-fourth of the protein amount than does ECNR2 expressing UAG-GFP (Figure 3C, 8.0 µg/ml for the GRO versus 35 µg/ml for ECNR2, p = 0.0014). […] These *ssrA* deletion strains likely demonstrate increased GFP expression and reduced growth rate and cell density (Figure 3A and 3B) because translation of GFP transcripts sequesters cellular resources at the expense of cellular replication, producing GFP peptides that are freed from nonstop ribosomes via ArfA or ArfB without addition of a degradation tag."

The reviewers are also correct to point out that, in Figure 3A, knockout of *arfB* does not seem to affect the induced/uninduced doubling time ratio. Indeed, ArfB does not seem to play a substantial role in influencing the growth rate of the GRO when it is expressing protein from UAG-ending transcripts. However, our data on protein expression represented in Figure 3C suggests that ArfB has a significant role to play in protein production. We emphasize this point on in subsection “Suppression, ribosomal frameshifting, and tmRNA-mediated peptide *ssrA* tagging occur at unassigned codons “.

d) The sentence in the last paragraph of subsection “ssrA and arfB mediate degradation of proteins containing unassigned UAG codons” is unclear: "Interestingly, a single knockout of arfB significantly reduced production of protein from UAG-GFP to low levels similar to those mapped to quantified GFP standards."

We thank the reviewers for pointing this out. Our intention is to demonstrate that UAG-GFP expression in the ∆*arfB* GRO strain leads to very low protein production levels (although there is no statistically significant difference in protein production between this strain and the GRO strain with no knockouts), and that the resulting protein abundance from this strain is near the lower limit of detection of our assay. We have revised subsection “Suppression, ribosomal frameshifting, and tmRNA-mediated peptide *ssrA* tagging occur at unassigned codons” to read: “A deletion of *arfB* leads to strikingly low protein abundances from UAG-GFP transcripts that approach the lower limit of detection of our assay, although this apparent reduction in protein production was not statistically significant in comparison to protein production by GRO.AA [pUAG-GFP].”

e) In subsection “ssrA and arfB mediate degradation of proteins containing unassigned UAG codons”, the comparison for "fully restore protein expression from UAG-ending transcripts" appears to be GRO.AA[pUAA-GFP] instead of ECNR2[pUAG-GFP], but "restore" suggests it should be the latter. The comparison and the meaning behind which strain is used for the comparison should be clarified.

The reviewer is correct, and we are indeed comparing protein production from GRO.AA.∆*ssrA*.∆*arfB* [pUAG-GFP] with GRO.AA [pUAA-GFP]. Because of the genetic differences between ECNR2 strains and GRO strains (Table 1), it would be inaccurate to compare protein production from GRO.AA.∆*ssrA*.∆*arfB* [pUAG-GFP] with that of ECNR2 [pUAG-GFP]. Our intention in making this comparison is to show that the knockout of both *ssrA* and *arfB* is needed to enable the GRO to translate peptides from UAG-ending transcripts at abundances similar to the translation of peptides from UAA-ending transcripts. We have revised subsection “*ssrA* and *arfB* mediate degradation of proteins containing unassigned UAG codons” to clarify the nature of our comparison, and it now reads: “These data also suggest that while deletion of *ssrA* partially recovers protein production from UAG-ending transcripts in the GRO, deletion of both *ssrA* and *arfB* is necessary to fully recover protein expression from UAG-ending transcripts to levels seen from the translation of UAA-ending transcripts in the GRO.”

f) In subsection “Deletion of ssrA restores conjugative plasmid propagation and viral infection in the GRO”, it is not clear what comparison was used for the 2.4-fold increase in doubling time. The graph in Figure 4B shows a 38% increase for the arfA strain versus 28% for the isogenic wild type.

We thank the reviewers for pointing this out. We revised the sentence in question to accurately reflect the data shown in the figure. Subsection “Deletion of *ssrA* restores conjugative plasmid propagation and viral infection in the GRO” now reads: “RK2 conjugation efficiency in GRO.AA.∆*ssrA* improved to 99% (compared to 87% in GRO.AA), and the strain showed an increase in doubling time of only 6% compared to a 28% increase for GRO.AA (p < 0.0001). We observed similar results for GRO.AA.∆*ssrA*.∆*arfB*. However, single deletion of *arfB* halved RK2 conjugative efficiency (Figure 4A, p = 0.0002). This strain also exhibited a 38% increase in doubling time when bearing RK2, compared to the 28% increase in doubling time seen in the GRO with no ribosomal rescue gene deletions (Figure 4B, p < 0.0001).”

g) Discussion section: What is the evidence supporting the demonstration of ribosome stalling? It seems that ribosome stalling was assumed based on the addition of the SsrA tag. Experiments such as ribosome profiling could demonstrate ribosome stalling, but these were not done. I think stalling is part of the model here but has not been demonstrated.

We appreciate the reviewer’s question and believe that clarifying the terminology used in the manuscript will resolve the issue. The term “ribosomal stalling” refers to the situation in which a ribosome reaches the 3’ end of an mRNA without encountering a stop codon, or when translation slows or pauses in the middle of an mRNA (Keiler, 2015). Stalling at the 3’ end of an mRNA results in ribosomal rescue (Keiler, 2015). Slowed translation within an mRNA can arise when the ribosome encounters a rare codon or a codon with depleted or inefficient cognate decoding elements, and can result in near-cognate suppression, frameshifting, or mRNA cleavage followed by ribosomal rescue (Aerni et al., 2015; George et al., 2016; Hayes et al., 2002; Keiler, 2015; Li et al., 2007; Roche and Sauer, 1999).

The reviewer is correct to point out that we do not present data to support the claim that unassigned codons result in ribosomal stalling as we have defined the term above. However, our mass spectrometry data does reveal that unassigned codons elicit translational responses (*i.e.,* ribosomal rescue, frameshifting, and near-cognate suppression) consistent with resolution mechanisms for stalled ribosomes.

We have modified the logic of the sentence in question to better reflect our experimental data. The sentence in the Discussion section now reads: “We demonstrate that unassigned stop codons elicit near-cognate suppression, frameshifting, and the action of ribosomal rescue mechanisms (Figure 2).” We conclude this paragraph with the additional sentence: “These mechanistic outcomes that occur as a consequence of ribosomal stalling could be further investigated via ribosomal profiling in future work.”

h) Discussion section: Similar to (g) above, the regulatory relationship between tmRNA and ArfA was used to explain the data, so it would be circular reasoning to then use this explanation to validate the regulatory relationship.

We thank the reviewers for pointing this out, and we agree that our data does not explicitly validate previously observed regulatory relationships between tmRNA and ArfA (as elucidated by Chadani et al., 2011; Garza-Sanchez et al., 2011; and Schaub et al., 2012). Our data does, however, reveal the physiological relevance of this relationship. In order to make this point more explicit, we have revised the sentence in the Discussion section to read: “Our GRO model thus sheds light on the functional significance of previously-described regulatory relationships while elucidating the unique mechanistic contributions of different ribosomal rescue systems in resolving translation at unassigned stop codons.”

i) Discussion section: "extensive" suggests there is a large amount of frameshifting, but the frameshifting events cannot be quantified using the techniques in this work. Perhaps something like "a wide variety of frameshifting events" would be more accurate.

We agree with the reviewer on this point. We have changed “extensive frameshifting” to “a wide variety of frameshifting events.”

j) Figure 1: The cartoon shows the SsrA-tagged protein going into the protease N terminus first, but all the proteases recognize the tag and start at the C terminus (this is not critical). In the legend, the word "hypothesized" should be removed from the last sentence – the lack of modification has been observed.

We thank the reviewers for this observation. We have modified the Figure to show the protease hydrolyzing the C-terminal end of the peptide and have removed the word “hypothesized” from the figure legend.

k) Figure 3 legend: For panels A and B, the legend does not match the labels. A is doubling time and B is max OD.

We thank the reviewers for pointing this out. We have altered the figure legend accordingly.